# Exercise facilitates post-stroke recovery through mitigation of neuronal hyperexcitability via interleukin-10 signaling

A. Schmidt-Pogoda[1] ✉, T. Ruck [2,3], JK Strecker[1,4], M. Hoppen[1], L. Fazio[5], L. Vinnenberg [5], B. Maus [6], L. Wachsmuth[6], M. Cerina[1], K. Diederich[1], S. Lichtenberg [4], H. Abberger [7,8,9], LAL Haertel[1], D. Schafflick [1,10], G. Meyer zu Hörste [1], AM Herrmann[5], P. Hundehege[1], V. Narayanan[1], C. Nelke[5], K. Kruithoff[1], J. Bosbach[1], E. Vicari[11], T. Ramcke[11], C. Beuker[1], E. Hadaschik[11], T. Budde [12], C. Faber [6], H. Wiendl [1,10], W. Hansen[7,13], SG Meuth [5,13] & J. Minnerup[1,4,13] ✉

Physical exercise is an effective therapy for improving stroke recovery. However, the exact underlying molecular mechanisms of exercise-enhanced neuronal repair remain unclear. As exercise affects the immune system in healthy individuals, and the immune system in turn influences recovery after stroke, we hypothesized that immune mechanisms play a role in exercise-induced neurological recovery. Using a model of ischemic stroke in adult male mice, we here show that the presence of regulatory T cells (Treg) within the ischemic brain is a prerequisite for exercise-enhanced functional and structural recovery. Treg prevent excessive and sustained hyperexcitability of periinfarct neurons via IL-10 signaling. This reduced hyperexcitability precedes alterations in neuronal connectivity, which underlie functional improvement. Together, we delineate the interaction of exercise-therapy, the immune system and functional recovery after ischemic stroke. Our findings can have translational relevance for further development of immune-targeted therapies.

Physical training is one of the most evidence-based neuroregenerative interventions in patients with stroke and has been shown to improve disability, mobility, and physical fitness, even when initiated days after symptom onset[1]. Consistent with findings from clinical trials, experimental stroke studies have also shown that physical exercise improves functional recovery after stroke[2]. At the cellular level, increased axonal sprouting, changes in cortical connectivity, and bihemispheric network reorganization have been identified as key structural changes associated with exercise-enhanced functional recovery[3,4]. Specifically, axonal sprouting has been observed in surviving pyramidal tract axons distal to the injury site and in the contralesional pyramidal tract, thus reinnervating deafferented corticospinal neurons[3,4]. However, the exact underlying molecular mechanisms, through which exercise facilitates neuroplasticity, are still unknown. It is believed that exercise

[1]Department of Neurology, University of Münster, Münster, Germany. [2]Department of Neurology, Ruhr University Bochum, BG University Hospital Bergmannsheil, Bochum, Germany. [3]Heimer Institute for Muscle Research, BG University Hospital Bergmannsheil, Bochum, Germany. [4]Department of Neurology, University of Lübeck and University Hospital Schleswig-Holstein, Lübeck, Germany. [5]Department of Neurology, Medical Faculty, Heinrich-Heine-University, Düsseldorf, Germany. [6]Clinic of Radiology, University of Münster, Münster, Germany. [7]Institute of Medical Microbiology, University Hospital Essen, University Duisburg-Essen, Essen, Germany. [8]Walter and Eliza Hall Institute of Medical Research, Parkville, VIC, Australia. [9]Department of Medical Biology, University of Melbourne, Parkville, VIC, Australia. [10]Department of Neurology and Neurophysiology, University Hospital Freiburg, Freiburg, Germany. [11]Department of Dermatology, Heidelberg University Hospital, Heidelberg, Germany. [12]Institute of Physiology I, University of Münster, Münster, Germany. [13]These authors jointly supervised this work: W. Hansen, SG Meuth, J. Minnerup. ✉e-mail: antje.schmidt-pogoda@ukmuenster.de; minnerup@uni-muenster.de

promotes the survival and growth of neurons via activation of signaling pathways and release of neurotrophic factors such as Brain-Derived Neurotrophic Factor (BDNF) and Insulin-like Growth Factor-1[5]. Additionally, exercise has well-established immunomodulatory effects, generating an overall anti-inflammatory environment[5,6]. Activation of the sympathetic nervous system, for instance, leads to an increased release of cortisol and adrenaline, which in turn inhibits the release of the proinflammatory cytokine tumor necrosis factor-α by monocytes[6]. Furthermore, exercise maintains an anti-inflammatory phenotype of adipose tissue, characterized by a high occurrence of macrophages associated with immune resolution and tissue repair, and regulatory T cells (Tregs), which release the anti-inflammatory cytokines Interleukin-10 (IL-10) and adiponectin[7,8]. Moreover, exercise mobilizes Tregs from lymphoid organs, thus increasing the number of circulating Tregs[9].

Considering the well-documented influence of the immune system on secondary damage and neuronal repair after stroke, we believe that the above-mentioned modulation of the immune system is a crucial mechanism through which exercise can attenuate postischemic damage and improve stroke recovery[10,11]. In particular, Tregs have been found to ameliorate postischemic damage by reducing the production of pro-inflammatory cytokines and suppressing neurotoxic astrogliosis[10,12,13].

At a molecular level, prior research has come to the significant insight that changes in neuronal excitability precede episodes of heightened neuroplasticity[14–16]. Since further studies have shown that T cells can influence neuronal excitability[17], we have put forward the intriguing hypothesis that exercise may also promote neuroplasticity through T cell-mediated modulation of neuronal excitability.

We here used a mouse model of ischemic stroke to analyze the relevance of different T cell subsets for the recovery-enhancing efficacy of exercise, T cell-dependent exercise-effects on neuronal network reorganization and exercise-effects on neuronal excitability. Our findings demonstrate that Tregs mediate exercise-enhanced stroke recovery via IL-10-signaling reduced neuronal hyperexcitability.

## Results

### Exercise improves the functional and structural recovery after stroke in wild type mice

We first analyzed the effect of exercise on stroke recovery in wild type mice (C57BL/6). To this end, wild type mice were subjected to photothrombotic stroke and then randomized into either exercise by forced running wheel training in motorized treadmills, starting 48 h after stroke, or no exercise. The adhesive tape removal test was employed for functional outcome assessment (Fig. 1a). Exercise significantly improved the functional outcome 49 days after stroke (Fig. 1b, *p = 0.04, t-test). Infarct volumes did not differ between both treatment groups (Fig. 1c, d, p = 0.16, t-test), which underlines that improved functional outcome after running wheel training is not a consequence of neuroprotection but is explained by truly regenerative effects of exercise. For histological quantification of axonal plasticity, we stereotactically injected the axonal tracers cascade blue (CB) and biotinylated dextranamine (BDA) into both motor cortices. These axonal tracers label axons within the pyramidal tract, so that axons can be visualized by immunohistochemistry. Forty-nine days after stroke, we quantified crossing fibers at the level of the facial nucleus, as previously described[18,19]. This anatomical region is particularly relevant for functional recovery, as the pyramidal tract at the level of the facial nucleus mediates voluntary motor control of the forelimbs and facial musculature—both of which are engaged during the adhesive tape removal test used in our study. Notably, mice often assist tape removal with their mouths, further emphasizing the functional importance of this area. The analysis at this level was therefore chosen to capture structural reorganization in a circuit directly linked to the behavioral improvements observed after training.

Forty-nine days after stroke, we found a significantly increased number of crossing fibers after running wheel training at the level of the facial nucleus (Fig. 1e–g, **p = 0.003 and *p = 0.04, t-test). These findings demonstrate that exercise enhances axonal plasticity after stroke. In accordance, magnetic resonance imaging (MRI)-diffusion tension imaging (DTI) fiber tracking, performed 49 days after stroke, revealed an increased number of interhemispheric connections in animals subjected to running wheel training (Fig. 1h–j, **p = 0.004, t-test). In summary, our experiments suggest that exercise improves functional recovery through structural repair after stroke.

### Regulatory T cells (Tregs) are prerequisites for exercise-enhanced recovery after stroke

We next aimed to clarify the importance of T cells and T cell-subsets for exercise-induced stroke recovery. To this end, we subjected RAG1$^{-/-}$-mice, which lack mature lymphocytes, to photothrombotic stroke and adoptively transferred different subsets of T cells (Fig. 2a). In all experiments, the adoptive T cell transfer was performed 24 h after stroke induction to target the post-acute phase—after the neuroprotective window in our model—thereby aiming to modulate regenerative processes without confounding effects on infarct development. An early post-stroke timing was chosen to avoid missing a critical window for initiating recovery. To substantiate this approach, we analyzed the temporal dynamics of Treg infiltration in exercised mice and found that Tregs were already present within the infarcted brain 3 days after stroke (Supplementary Fig. 1). This observation supports the rationale for performing the adoptive transfer prior to this time point to effectively influence Treg-mediated regeneration.

In the initial experiments aiming to evaluate which T cell subtype is relevant for the observed effects, we administered 3 million total T cells per animal intraperitoneally 24 h after ischemia induction. For Treg transfer experiments, we injected 500,000 Tregs per animal.

Most interestingly, exercise did not improve the recovery of RAG1$^{-/-}$-mice without adoptive T cell-transfer (Fig. 2b, p = 0.46, t-test). Consistently, adoptive transfer of CD3$^+$ T cells alone, in the absence of exercise, was also insufficient to enhance recovery in RAG1$^{-/-}$-mice (Fig. 2c, p = 0.28, t-test), indicating that T cells require an exercise context to mediate functional improvement.

Following an adoptive transfer of CD3$^+$-T cells, exercise improved the recovery of RAG1$^{-/-}$-mice (Fig. 2d, ***p = 0.0002, t-test). This key finding points out that T cells are required to enable exercise-induced stroke recovery. We next aimed to identify the subset of T cells that drive stroke recovery. For this purpose, we transferred either CD4$^+$- or CD8$^+$-T cells to RAG$^{-/-}$-mice and compared the efficacy of exercise. Our results show that CD4$^+$-T cells rather than CD8$^+$-T cell are required for exercise-enhanced improvement of functional outcomes (Fig. 2e, p = 0.07, t-test). We further differentiated between FoxP3$^+$ regulatory T cells (Tregs) and FoxP3$^-$ nonregulatory T cells and found that Tregs represent the subset of T cells that drive functional recovery (Fig. 2f, ***p = 0.0009, t-test). To distinguish between central and peripheral immune cell effects, we applied an antibody against very late antigen-4 (VLA-4), starting 24 h after stroke induction, to prevent leukocytes from entering the brain. Following VLA-4-blockade, animals had a significantly impaired functional recovery compared to animals that received an isotype antibody (Fig. 2g, *p = 0.02, t-test), thus indicating that central immune cell-effects are required to enable exercise-enhanced stroke recovery. To clarify the importance of Tregs for exercise-induced structural repair, we subjected RAG1$^{-/-}$-mice to photo-thrombotic stroke, adoptive transfer of Tregs or vehicle, exercise and MRI-DTI fiber tracking. In accordance with the above finding that Tregs are promotors of functional improvement after stroke, MRI-DTI fiber tracking performed 49 days after stroke revealed an increased number of interhemispheric connections in mice that

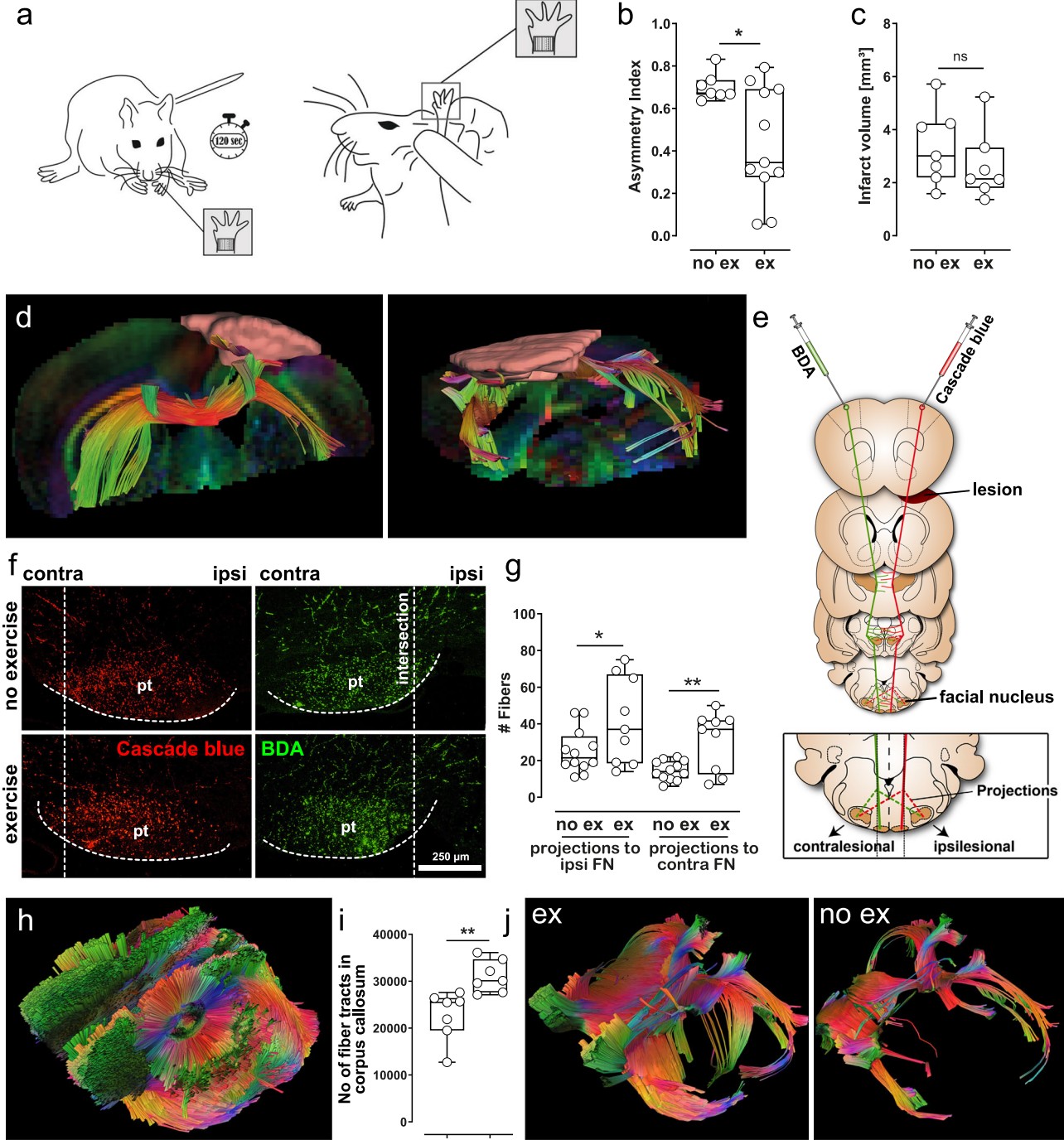

**Fig. 1 | Exercise improves the functional and structural recovery after stroke in wild type mice. a** The adhesive tape removal test was used for functional outcome assessment. The removal index is calculated as follows: (removal time of impaired forelimb−removal time of unimpaired forelimb)/(removal time of impaired forelimb + removal time of unimpaired forelimb). **b** Exercise significantly reduced the asymmetry score in the adhesive tape removal test 49 days after photothrombotic stroke (*$p$ = 0.04, two-sided $t$-test, $n$ = 12 and 8 animals per group), which indicates improved functional outcome (ex = exercise, no ex = no exercise). **c** Mean infarct volumes were 2.64 mm³ ± 0.33 mm³ and 3.35 mm³ ± 0.36 mm³ ($p$ = 0.16, two-sided $t$-test, $n$ = 7 animals per group). **d** Exemplary coronary and sagittal MRI scans illustrate infarct size and location. **e** Axonal tracers CB and BDA were stereotactically injected into both motor cortices thus anterogradely labeling axons of pyramidal tract axons. **f** CB- and BDA-labeled axons were visualized immunohistochemically and crossing fibers were quantified at the level of the facial nucleus. **g** Running

wheel training increased the number of crossing fibers, both ipsilaterally and contralaterally (**$p$ = 0.003 and *$p$ = 0.04, two-sided $t$-test, $n$ = 9 and 12 animals per group), suggesting that exercise enhances axonal plasticity after stroke. **h** To better characterize exercise effects on neuronal network reorganization, we employed MRI-DTI fiber tracking (Color code: Fiber orientation: green: dorso-ventral, red: left-right, blue: rostro-caudal). **i** MRI-DTI fiber tracking demonstrated an increased number of interhemispheric connections after running wheel training (**$p$ = 0.0043, two-sided $t$-test, $n$ = 7 animals per group). **j** Exemplary scans of animals with a high (ex) and a low (no ex) number of interhemispheric connections. For box and whisker plots, the box extends from the 25th to the 75th percentile, the center is the median, and whiskers extend from the minimum or maximum. Each dot represents an individual biological replicate. Asterisks indicate levels of statistical significance: *$p$ < 0.05, **$p$ < 0.01. Source data are provided as a Source Data file.

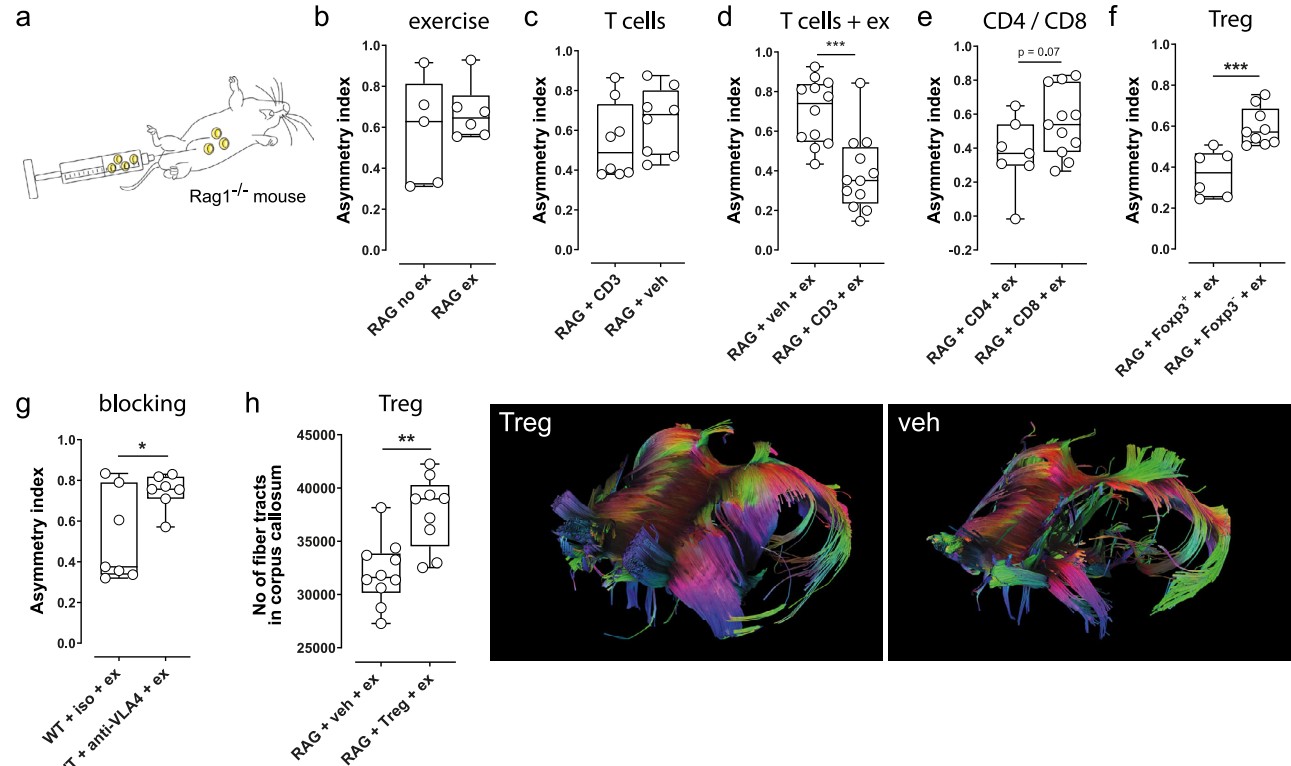

**Fig. 2 | Regulatory T cells (Tregs) are promotors of exercise-enhanced functional and structural recovery after stroke. a** RAG1$^{-/-}$-mice were subjected to photothrombotic stroke, received different subsets of T cells or vehicle (veh), and were then subjected to running wheel training. **b** Running wheel training did not influence the asymmetry index in the adhesive tape removal test ($p = 0.46$, two-sided $t$-test, $n = 6$ and 5 animals per group) in RAG1$^{-/-}$-mice without adoptive T cell-transfer (ex = exercise, no ex = no exercise). **c** Adoptive transfer of CD3$^+$ T cells, in the absence of exercise, did not enhance recovery in RAG1$^{-/-}$ mice ($p = 0.28$, two-sided $t$-test, $n = 8$ and 8 animals per group). **d** Following adoptive CD3$^+$-T cells-transfer, exercise led to a significantly reduced asymmetry index (***$p = 0.0002$, two-sided $t$-test, $n = 13$ and 12 animals per group), indicating improved functional outcome. **e** There was a strong trend towards better functional outcome after CD4$^+$- compared to CD8$^+$-T cell-transfer ($p = 0.07$, two-sided $t$-test, $n = 7$ and 11 animals per group). **f** The transfer of FoxP3$^+$ regulatory T cells (Tregs) resulted in significantly

better functional outcomes than the transfer of FoxP3$^-$ nonregulatory T cells (***$p = 0.0009$, two-sided $t$-test, $n = 6$ and 10 animals per group). **g** Preventing leukocytes from entering the brain by inhibition of very late antigen-4 (VLA-4) impaired stroke recovery (*$p = 0.02$, two-sided $t$-test, $n = 7$ animals per group; iso = isotype antibody). **h** MRI-DTI fiber tracking demonstrated an increased number of interhemispheric connections in mice that received Tregs before running wheel training compared to mice that received only vehicle and running wheel training (**$p = 0.0013$, two-sided $t$-test, $n = 9$ and 10 animals per group). Exemplary scans of animals with a high (ex) and a low (no ex) number of interhemispheric connections. For box and whisker plots, the box extends from the 25th to the 75th percentile, the center is the median and whiskers extend from the minimum or maximum. Each dot represents an individual biological replicate. Asterisks indicate levels of statistical significance: *$p < 0.05$, **$p < 0.01$, ***$p < 0.001$. Source data are provided as a Source Data file.

received Tregs compared to mice that received vehicle (Fig. 2h, **$p = 0.001$, $t$-test).

Infarct volumes were comparable between wild-type mice with stroke and exercise treated with either anti-VLA-4 antibody or isotype control, as well as between RAG1$^{-/-}$-mice with stroke receiving Treg transfer and exercise or vehicle transfer and exercise (Supplementary Fig. 2). These findings support the conclusion that the observed improvements in functional recovery are attributable to regenerative mechanisms rather than acute neuroprotective effects.

Altogether, these findings highlight that the presence of Tregs in the ischemic brain is a prerequisite for exercise-enhanced recovery after stroke.

### Exercise increases the proliferation of Tregs in the ischemic brain

We next aimed to characterize exercise-effects on T cell subsets in the ischemic brain, cervical lymph nodes (cLN) and spleen 14 days after stroke. Our results show that exercise resulted in a significant reduction of brain-infiltrating CD45$^{high}$ immune cells and CD3$^+$ T cells (Fig. 3a). We did not observe any effect on relative cell counts of CD4$^+$- and CD8$^+$- T cells among CD3$^+$ T cells (Fig. 3b), but found a tendency towards an increased proportion of FoxP3$^+$CD4$^+$-regulatory T cells (Tregs) in the

brain ($p = 0.1$, $t$-test, Fig. 3c) and an increased proportion of Tregs in cervical lymph nodes (*$p = 0.01$, Mann-Whitney-test, Fig. 3c). The expression of integrin CD49d, which facilitates leukocyte-trafficking into the brain, was upregulated in CD4$^+$-T cells after exercise (*$p = 0.03$, Mann-Whitney-test, Fig. 3d). Interestingly, exercise resulted in significantly enhanced proliferation of FoxP3$^+$CD4$^+$ Tregs in the brain (*$p = 0.04$, $t$-test, Fig. 3e), whereas the proliferation of FoxP3$^-$CD4$^+$ and CD8$^+$ effector T cells was impaired both in the brain (*$p = 0.03$, $t$-test) and cervical lymph nodes (cLN) (*$p = 0.03$; **$p = 0.006$, $t$-test, Fig. 3e) as determined by Ki67 expression. In additional time-course experiments, we further demonstrated that Tregs infiltrate the infarcted brain early after stroke and are already detectable within the lesion area as early as 72 h post-stroke, with levels remaining relatively stable throughout the first week (Supplementary Fig. 1).

In summary, exercise after stroke dampens the proliferation of effector T cells and elevates the proliferative capacity of Tregs within the ischemic brain.

### Exercise induces a shift towards anti-inflammatory immune cell polarization

Heat maps of immigrating CD3$^+$-T cells did not reveal any apparent differences in their spatial distribution at 14 days or 49 days post stroke

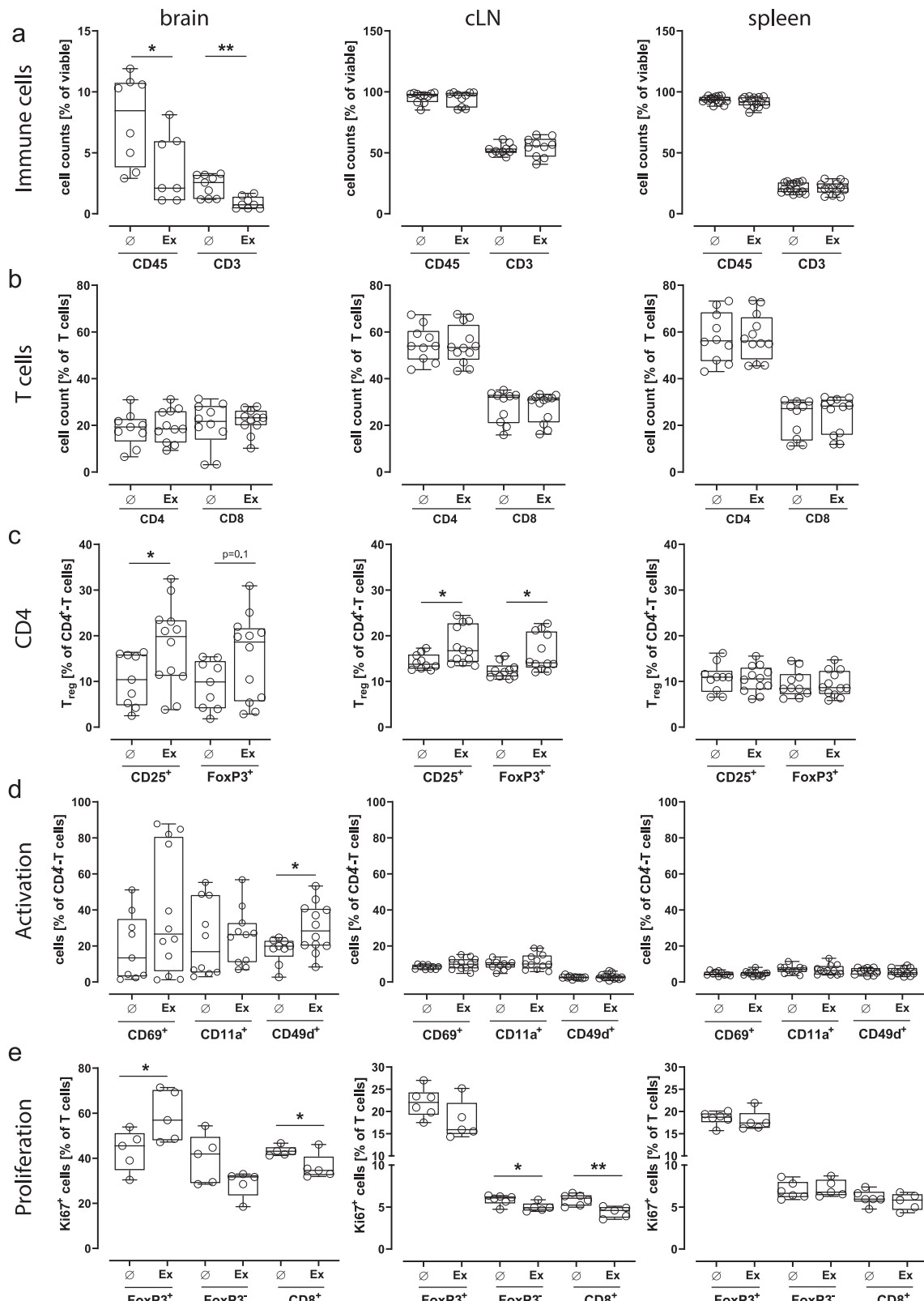

(Fig. 4a). Interestingly, immunohistochemical quantification of CD3+-T cells showed increased numbers of T cells in the ipsilateral thalamus and in the ipsilateral internal capsule in the recovery phase at 49 days after stroke (Fig. 4a, *p = 0.01 and p = 0.05, t-test). Figure 4b–d illustrate exemplary immunohistochemical stainings of CD3+-T cells in the infarct core (Fig. 4b) and in the internal capsule (Fig. 4c, d). To visualize FoxP3+-regulatory T cells in the brain parenchyma, we employed

genetically modified FoxP3+-mRFP-mice, in which FoxP3+-cells express the red fluorescent protein and can thus be detected by fluorescence microscopy (Fig. 4e). Visual inspection of heat maps did not reveal apparent differences in the spatial distribution of immigrating Tregs (Fig. 4e).

To better characterize the effects of exercise on the immune response we performed RT2 profiler real time polymerase chain

**Fig. 3 | Exercise increases the proportion of Tregs in brain and cervical lymph nodes. a** Quantification of immune cells by flow cytometry 14 days after stroke revealed that exercise significantly reduced brain-infiltrating CD45$^{high}$ immune cells and CD3$^+$ T cells (*$p$ = 0.04 and **$p$ = 0.002, two-sided $t$-test, $n$ = 7–9 animals per group), but had no impact on the count of CD45$^{high}$ immune cells and CD3$^+$ T cells in cervical lymph nodes (cLN) and spleen. **b** Exercise did not influence the proportions of CD4$^+$- and CD8$^+$-T cells in brain parenchyma, cervical lymph nodes and spleen 14 days after photothrombotic stroke (two-sided $t$-test, $n$ = 9–12 animals per group). **c** Exercise increased the proportion of CD25$^+$CD4$^+$-and FoxP3$^+$CD4$^+$-regulatory T cells in brain parenchyma (*$p$ = 0.04 and $p$ = 0.1, two-sided $t$-test, $n$ = 9–12 animals per group) and cervical lymph nodes (*$p$ = 0.01, Mann-Whitney-test and **$p$ = 0.01, two-sided $t$-test, $n$ = 9–12 animals per group). **d** The expression of integrin CD49d, which facilitates leukocyte-trafficking into the brain, was upregulated in brain CD4$^+$-T cells after exercise (*$p$ = 0.03, Mann-Whitney-test,

$n$ = 9–12 animals per group), but not in CD4$^+$-T cells isolated from cervical lymph nodes or spleen. The expression of the activation markers CD69 and CD11a on CD4$^+$-T cells did not differ in any of the examined compartments. **e** Exercise led to a significant increase in the proliferation of FoxP3 + CD4+ regulatory T cells (Tregs) in the brain (*$p$ = 0.04, two-sided $t$-test, $n$ = 4–5 animals per group). Conversely, the proliferation of FoxP3-CD4$^+$ and CD8$^+$ effector T cells exhibited impairment both in the brain (*$p$ = 0.03, two-sided $t$-test, $n$ = 4–5 animals per group) and cervical lymph nodes (cLN) (*$p$ = 0.03; **$p$ = 0.006, two-sided $t$-test, $n$ = 5–6 animals per group) as determined by Ki67 expression. For box and whisker plots, the box extends from the 25th to the 75th percentile, the center is the median and whiskers extend from the minimum or maximum. Each dot represents an individual biological replicate. Asterisks indicate levels of statistical significance: *$p$ < 0.05, **$p$ < 0.01. Source data are provided as a Source Data file.

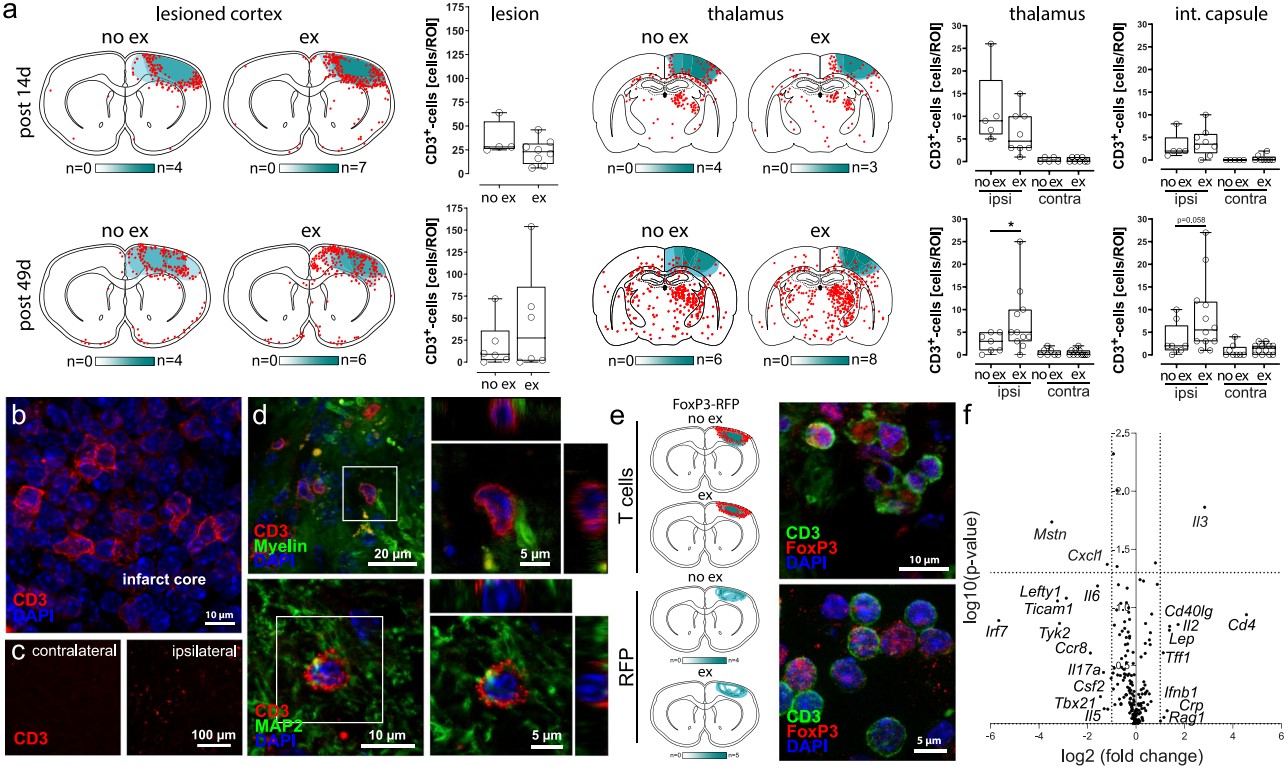

**Fig. 4 | Exercise induces a shift towards anti-inflammatory immune cell polarization. a** Heatmaps based on immunohistochemical CD3-staining did not reveal apparent differences in the spatial distribution of immigrating CD3$^+$-T cells between trained and untrained animals. Quantification of CD3$^+$-T cells in different areas of interest in lesioned cortex, Thalamus and internal capsule 14 or 49 days after ischemia did not show differences in the cell count (two-sided $t$-test, $n$ = 4–8 animals per group). **b–d** Representative images of immunohistochemical stainings from the infarct area and the internal capsule; similar results were observed in all animals analyzed ($n$ = 28 across all groups). **e** To visualize immigrating FoxP3$^+$-regulatory T cells in the brain parenchyma, we employed genetically modified FoxP3$^+$-RFP-mice, in which FoxP3$^+$-cells are labeled with the red fluorescent protein and can thus be detected by fluorescence microscopy. Heatmaps based on

fluorescence microscopy did not reveal apparent exercise effects on the spatial distribution of immigrating FoxP3$^+$-regulatory T cells in the brain parenchyma ($n$ = 4 and 5 animals per group). **f** Volcano plot analysis of RT² profiler displays differential gene expression in mice with or without exercise 14 days after experimental stroke. Each data point represents a transcript, with the abscissa displaying log2 fold change and the ordinate statistical significance (negative log10 $p$ value). Data are generated from 8 wildtype mice (no exercise $n$ = 4; exercise $n$ = 4, two-sided $t$-test). For box and whisker plots, the box extends from the 25th to the 75th percentile, the center is the median and whiskers extend from the minimum or maximum. Each dot represents an individual biological replicate. Asterisks indicate levels of statistical significance: *$p$ < 0.05. Source data are provided as a Source Data file.

reaction (PCR) arrays of brain lysate from the brain 14 days after stroke.

RT2 profiler real time PCR showed a significantly reduced expression of the chemokine CXCL1, which is crucial in inflammation and acts as a chemoattractant for neutrophils, in animals that underwent exercise (Fig. 4f, *$p$ = 0.04). Further, we found a strong tendency towards reduced pro-inflammatory cytokine IL-6 (Fig. 4f, $p$ = 0.06), which is known as a key player in stroke pathophysiology with predominantly harmful effects. Interleukin-3, which acts as a

growth factor for hematopoietic stem cells and promotes immune cell differentiation, was most clearly upregulated (Fig. 4f, $p$ = 0.0014). Hence, running wheel training after stroke induces a shift towards a less inflammatory local cytokine-expression profile within the ischemic brain.

## Exercise suppresses neurotoxic astrogliosis

A prior study suggested that Tregs enhance neurological recovery in chronic stroke by suppressing neurotoxic astrogliosis[12]. Since our

findings demonstrate increased proliferation of Tregs in the ischemic brain after exercise, we aimed to clarify whether exercise reduces astrogliosis and thereby contributes to improved stroke recovery. As microglial activation can influence the phenotype of astrocytes and induce a switch towards a so called A1-phenotype with increased release of proinflammatory cytokines[20–22], we also investigated microglial activation. Our results showed that exercise indeed reduced astrogliosis in the lesioned cortex and the ipsilateral thalamus 14 days post stroke Fig. 5a–d, *$p = 0.04$, t-test, and $p = 0.05$, t-test). Concomitantly, we also found decreased numbers of microglia/macrophages in the lesioned cortex and the contralateral thalamus 14 days after stroke (Fig. 5f, *$p = 0.02$, t-test, and *$p = 0.02$, Mann-Whitney-test). In the chronic stroke phase after 49 days, when microglia and macrophages are thought to contribute to tissue remodeling[23] and regeneration exercise led to an increased presence of these cells in the lesioned cortex (Fig. 5e, *$p = 0.04$, t-test), whereas their numbers were reduced in the contralateral cortex (Fig. 5e, *$p = 0.04$, t-test). Notably, our findings suggest that exercise promotes a shift towards a phenotype associated with tissue repair and anti-inflammatory functions[23]: Fourteen days after stroke, we found increased numbers of cells expressing Arginase-1, a marker commonly linked to repair-associated macrophage responses (Fig. 5g, h, *$p = 0.02$, t-test). By day 49 post-stroke, a strong trend towards an increase in CD206-expressing cells was observed, which is indicative of macrophage involvement in tissue remodeling and homeostasis (*Fig. 5g, h, $p = 0.07$, t-test).

In summary, we demonstrate that exercise suppresses potentially neurotoxic astrogliosis and reduces microglial activation in the early phase of stroke recovery. Furthermore, our findings support the idea that exercise influences macrophage polarization towards a phenotype associated with repair and resolution of inflammation[23].

## Exercise modulates neuronal activity Treg-dependently

Previous studies showed that cortical strokes induce abnormal excitability of periinfarct neurons and changes in excitability may precede heightened neuroplasticity during the early phase of stroke recovery[14–16]. As T cells are shown to influence neuronal excitability and survival[17,24], we aimed to investigate T cell-dependent exercise-effects on periinfarct-excitability. In our baseline experiments, we challenged neurons in the periinfarct area of untreated animals with a depolarizing current pulse (+160 pA) using the whole-cell patch clamp technique and subsequently quantified action potentials (Fig. 6a). Most interestingly, we found a hyperexcitability of periinfarct neurons compared to the sham operated controls 14 days after stroke (Fig. 6b, $p = 0.05$, Mann-Whitney-test), thus indicating that stroke induced a hyperexcitability of periinfarct neurons. To analyze exercise-effects on neuronal activity after stroke, we compared periinfarct excitability in exercise mice and control mice. Our results showed that exercise normalized the post-stroke hyperexcitability in wild type mice (Fig. 6c, *$p = 0.013$, t-test). No exercise-related differences in action potential generation were observed in sham-operated animals (Fig. 6d, $p = 0.3$, Mann–Whitney-test). Likewise, in RAG1$^{-/-}$-mice, exercise had no effect on the excitability of periinfarct neurons (Fig. 6e, $p = 0.8$, t-test). However, when RAG1$^{-/-}$-mice received Treg transfer, exercise was associated with a strong trend towards reduced neuronal excitability in the periinfarct cortex, indicating a potential Treg-dependent effect of exercise on neuronal activity (Fig. 6f, $p = 0.07$, t-test).

To confirm that the immune cell-dependent effects of exercise on neuronal excitability are specific to the stroke context, we performed additional patch-clamp recordings in sham-operated wild-type animals. In these experiments, blockade of leukocyte–endothelial interactions using a VLA-4 antibody did not affect cortical neuron excitability in exercised sham animals (Supplementary Fig. 4a). These results further substantiate the specificity of the observed effects to

the post-stroke condition. Given the limited sample size in these control experiments, the absence of significant changes should be interpreted with some caution, though. Comparison of wild-type and RAG1$^{-/-}$-mice revealed no significant difference in neuronal excitability on the ipsilateral side (Supplementary Fig. 4b), but a significant genotype effect on the contralateral side (Supplementary Fig. 4c); however, since identical genotypes were compared within each experimental group, we do not consider this finding to critically impact our conclusions.

Overall, our data indicate that exercise improves recovery and neuronal network reorganization by modulating neuronal activity Treg-dependently.

## Interleukin-10 is a key mediator of exercise-induced stroke recovery

Finally, we aimed to identify the pathway by which Tregs mediate exercise-induced stroke recovery. In the first place, we wanted to rule out the possibility that the mere presence of activated CD4$^+$ T cells as a cofactor is sufficient to enable exercise-induced stroke recovery and a mediating pathway does simply not exist. For this purpose, we transferred either dysfunctional FoxP3-deficient CD25$^+$CD4$^+$ T cells from scurfy mice or vehicle to RAG1$^{-/-}$-mice and analyzed exercise-effects on the functional recovery (Fig. 7a). Our results show that the transfer of dysfunctional CD25$^+$CD4$^+$ T cells did not improve the functional recovery compared to vehicle (Fig. 7b, $p = 0.29$, t-test), thus demonstrating that the mere presence of activated CD4$^+$ T cells as a cofactor is not sufficient to enable exercise-induced stroke recovery. Considering that IL-10 is one of the most potent pro-regenerative cytokines secreted by Tregs, we hypothesized that IL-10 is crucial for exercise-enhanced stroke recovery. To verify this hypothesis, we transferred Tregs from Interleukin-10-deficient (IL-10$^{-/-}$) - or wild type mice and analyzed exercise-effects on the functional and structural recovery. As the IL-10$^{-/-}$- mice lack a labeled FoxP3, and given that FoxP3 is a nuclear marker, FACS sorting of live FoxP3$^+$ cells was not feasible. Therefore, for the subsequent transfer experiments, CD4$^+$CD25$^+$ T cells were considered as Tregs. To ensure the closest possible match to the FoxP3$^+$ Treg population, we conducted co-staining of glucocorticoid-induced tumor necrosis factor-related receptor (GITR) and CD304 as a quality control measure. Indeed, more than 90% of CD4$^+$CD25$^+$ T cells from WT mice as well as from IL-10-deficient mice express FoxP3 (Supplementary Fig. 5a) and both Treg populations are able to inhibit the proliferation of anti-CD3 stimulated CD4$^+$CD25$^-$ T cells to a similar extent in vitro (Supplementary Fig. 5b). Our results showed that the transfer of IL-10$^{-/-}$-Tregs resulted in significantly worse functional recovery compared to the transfer of Tregs from wildtype mice (Fig. 7c, ** $p = 0.003$, t-test). On a structural level, MRI-DTI fiber tracking showed a lower number of interhemispheric connections in mice that received IL-10$^{-/-}$-Tregs compared to mice that received Tregs (Fig. 7d, e, *$p = 0.03$, t-test). As our electrophysiological experiments prompted us to assume that exercise improves neuronal network reorganization by modulating neuronal activity Treg-dependently, we next aimed to clarify, if IL-10 is also crucial for Treg-effects on neuronal activity. To this end, we transferred either IL-10$^{-/-}$-Tregs or wildtype Tregs to RAG1$^{-/-}$-mice and assessed exercise-effects on neuronal activity of periinfarct neurons. Most importantly, our experiments showed that, in contrast to wildtype Tregs, IL-10$^{-/-}$-Tregs did not enable exercise-mediated normalization of neuronal hyperexcitability (Fig. 7f, g, ** $p = 0.002$, t-test). To verify that IL-10-deficient Tregs do not affect neuronal excitability independently of stroke, we performed additional patch-clamp recordings in RAG1$^{-/-}$ sham animals receiving IL-10$^{-/-}$ Tregs. These recordings revealed no significant changes in neuronal excitability compared to untreated sham controls (Supplementary Fig. 4d).

These findings confirm IL-10 as a key mediator for exercise-enhanced recovery after stroke.

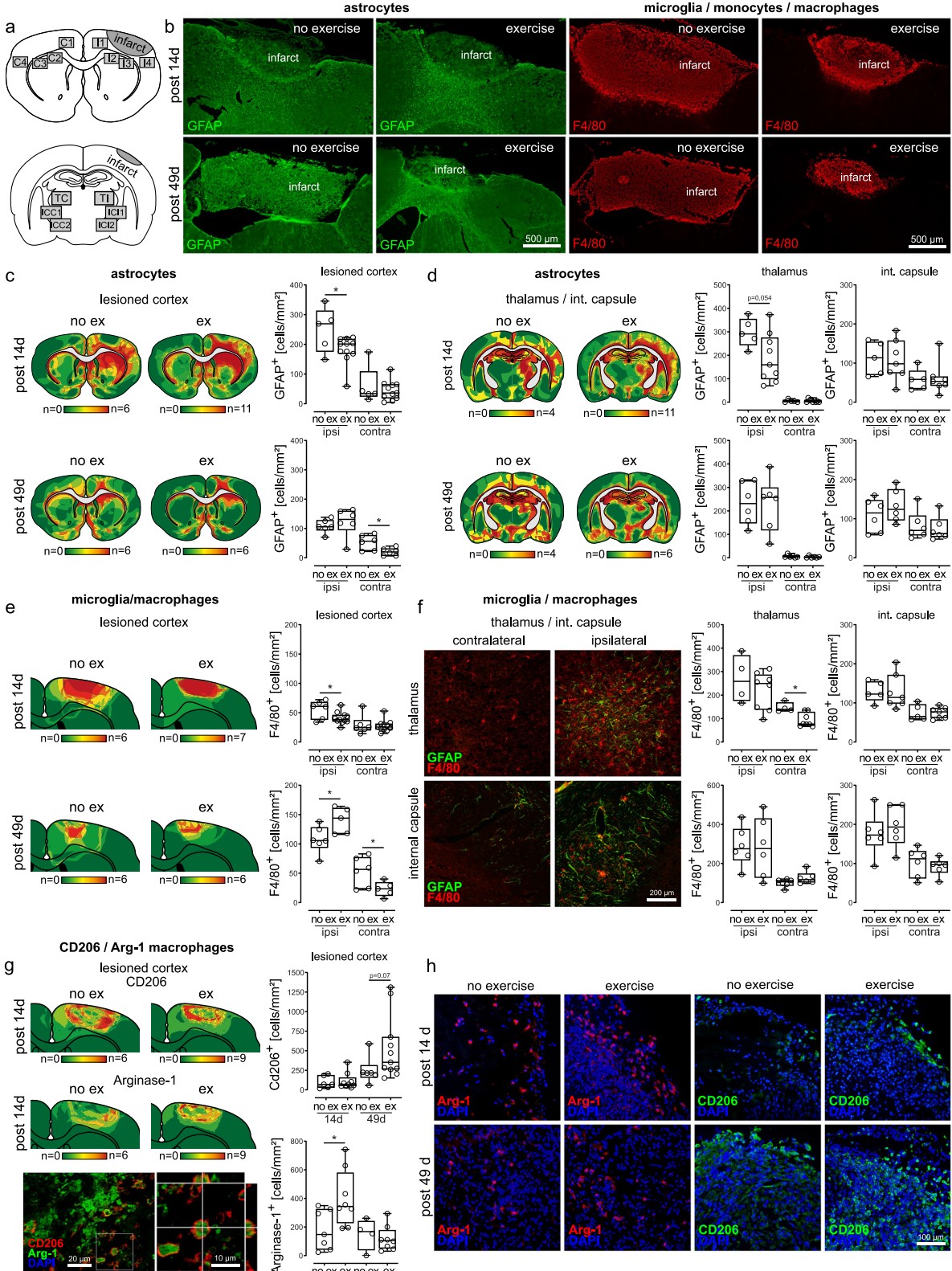

## Discussion

Our experiments identify IL-10-signaling by Tregs as a prerequisite for exercise-enhanced stroke recovery. Mechanistically, we show that IL-10 modulates the excitability of periinfarct neurons, which precedes heightened neuroplasticity[14,15,25]. In our experiments, heightened neuroplasticity with increased neuronal network reorganization becomes evident by augmented axonal sprouting of pyramidal tract neurons

and augmented interhemispheric connectivity visualized by MRI-DTI fiber tracking.

In animals that had not undergone exercise, we observed a persisting hyperexcitability of periinfarct neurons in response to electrical stimulation for at least 14 days after stroke. Such maladaptive cortical hyperexcitability induced neurodegeneration by increased caspase-3 expression and upregulation of TNF-α in excitatory neurons in a model

**Fig. 5 | Exercise suppresses astrogliosis and enhances macrophage expression of markers associated with tissue repair. a** Astrocytes and microglia/macrophages were quantified in regions of interest cortex, Thalamus and internal capsule. **b** Representative immunohistochemical stainings of the glial fibrillary acidic protein (GFAP) and surface protein F4/40, performed to visualize astrocytes and microglia/macrophages; similar results were observed in all animals analyzed (n = 29 across all groups). **c, d** Exercise reduced astrogliosis in the lesioned cortex and the ipsilateral Thalamus 14 days post stroke (*p = 0.04 and p = 0.05, two-sided t-test, n = 4–11 animals per group). **e, f** Exercise led to decreased numbers of microglia/macrophages in the lesioned cortex and the contralateral Thalamus 14 days after stroke (*p = 0.02, two-sided t-test, and *p = 0.02, Mann-Whitney-test, n = 6-7 animals per group). In the chronic stroke phase after 49 days, exercise resulted in an increased cell count of microglia/macrophages in the lesioned cortex

(*p = 0.04, two-sided t-test, n = 6–7 animals per group), whereas their numbers were decreased in the contralateral cortex (*p = 0.04, two-sided t-test, n = 6–7 animals per group). **g, h** Fourteen days after stroke, we found increased numbers of cells expressing Arginase-1, a marker commonly linked to repair-associated macrophage responses (*p = 0.02, two-sided t-test, n = 6–9 animals per group). Forty-nine days post-stroke, we observed a rise in the quantity of cells expressing CD206, indicative of a macrophage phenotype associated with tissue remodeling and homeostasis (*p = 0.02, t-test). For box and whisker plots, the box extends from the 25th to the 75th percentile, the center is the median and whiskers extend from the minimum or maximum. Each dot represents an individual biological replicate. Asterisks indicate levels of statistical significance: *p < 0.05. Source data are provided as a Source Data file.

of experimental autoimmune encephalitis[26]. Cortical hyperactivity was paralleled by behavioral disturbances, whereas a normalization of neuronal network activity patterns via TNF-α-blockade alleviated behavioral abnormalities[26]. Others confirmed a strong link between dysfunctional adaptive immunity, disturbed neuronal network activity patterns and abnormal behavior[17,27,28]. In particular, altered T cell-responses were associated with dysregulated connectivity and social deficits[17,27,28]. For instance, mice deficient in meningeal T cells exhibited behavioral disturbances and a hyper-connectivity of fronto-cortical brain regions, whereas IFN-γ derived from meningeal T cells elevated tonic GABAergic inhibition and prevented aberrant excitability[17].

In our experiments, exercise normalized periinfarct hyperexcitability via IL-10-signaling by Tregs. Likewise, anti-excitatory actions of IL-10 were previously observed in models of pain and hypoxia[29–31].

Following stroke, the periinfarct area is critical for recovery, because it can either allow remapping and repair through raised neuroplasticity or contribute to further loss of function by neurodegenerative processes[32]. Electrophysiologically, cortical infarcts induce a transient hypoexcitability of adjacent periinfarct neurons due to excessive GABA-mediated tonic inhibition[14]. Antagonizing this excessive GABA-mediated tonic inhibition by a benzodiazepine inverse agonist led to an early and sustained improvement of functional recovery[14]. Similarly, reducing excessive GABAergic activity facilitated neuronal plasticity in genetic models of cognitive disorders[33,34]. Our electrophysiological data indicate that exercise normalizes stroke-induced hyperexcitability rather than reducing neuronal excitability below physiological levels. This normalization may prevent excitotoxicity and support structural remodeling, aligning with current concepts of post-stroke neuroplasticity that emphasize the importance of a tightly regulated balance of excitability to enable neuroplasticity[32]. While our data support an IL-10-dependent mechanism modulating neuronal excitability, we cannot exclude the possibility that IL-10 may act indirectly throuhg cell types other than neurons. Future studies employing neuron-specific IL-10 receptor deletion or blockade will be required to confirm a direct effect of IL-10 on neurons. Taking into account our own findings and the above-mentioned findings from others, we conclude that an initial increase of neuronal activity in the early phase after stroke enhances plasticity and repair, whereas persisting hyperexcitability induces neurodegeneration. In other words, a well-balanced, previously described "Yin and Yang"[32] of periinfarct excitability creates an environment that fosters stroke recovery.

Apart from influencing neuronal activity, IL-10 may prevent secondary injury after stroke by limiting the production of proinflammatory cytokines, TNF-α, IFN-γ, IL-1β and IL-17, restricting neutrophil infiltration and modulating microglial activation and lymphocyte invasion[10,35]. In a previous study, both, an early intracerebroventricular administration of IL-10 and an adoptive Treg-transfer reduced infarct sizes in lymphocyte deficient mice, whereas an adoptive transfer of Tregs obtained from IL-10-deficient mice did not[10]. This highlights the key importance of IL-10-signaling and is in accordance

with our own adoptive transfer experiments which showed that the transfer of Tregs obtained from IL-10-deficient mice did not enable exercise-enhanced stroke recovery and neuronal network reorganization.

Besides IL-10-mediated pathways, Tregs may have improved stroke recovery by suppressing neurotoxic astrogliosis, as recently described[12]. Depending on the mechanism of CNS injury and the surrounding milieu, there are at least two opposing types of reactive astrocytes, which can either promote or hinder recovery[20–22]. In response to reactive microglia with increased release of proinflammatory cytokines such as TNF-α, astrocytes switch to a so called A1-phenotype hindering repair and inducing death of neurons and oligodendrocytes[20–22]. In our experiments, the suppression of astrogliosis in exercised mice was accompanied by a reduction in microglial activation and a shift in microglia/macrophage phenotypes towards a state associated with tissue repair and inflammation resolution[23]. This adaptive immune response may represent an additional mechanism through which exercise facilitated neuronal repair.

While the number of transferred Tregs detected in the peri-infarct region was relatively low, this does not preclude the possibility that peripheral immunomodulatory effects may also contribute to stroke recovery. We therefore acknowledge that both central and systemic mechanisms may act in concert to mediate the observed benefits of adoptively transferred Tregs.

A major strength of our study is that we found improved outcomes both on a functional level as indicated by positive results in behavioral tests and on a structural level as demonstrated by MRI and histology. As an underlying mechanism, we identified changes in the excitability of peri-nfarct neurons mediated by IL-10-release from Tregs. Our study is limited by the use of a single behavioral test to assess functional recovery, which may not capture all aspects of sensorimotor performance. However, the adhesive tape removal test has proven to be particularly sensitive and robust in our stroke model[36,37] and aligns well with the anatomical and functional effects observed in our study. A caveat is that our immune cell analyses do not indicate increased levels of IL-10 in the ischemic brain. However, our FACS analyses 14 days after stroke show increased proliferative capacity of Tregs in the ischemic brain of exercise mice and IL-10 represents one of the most significant Treg-produced cytokines in stroke regeneration, so it stands to reason that Tregs also released IL-10 in our experiments. Further, our analyses were performed 14 days after stroke, while others observed that Tregs migrate into the brain primarily within the first week after stroke[10]. Thus, it is quite conceivable that there was an even higher number of Il-10-producing Tregs in the brain 1 week after stroke than 14 days after stroke and that IL-10 concentrations were indeed elevated within the first week in our experiments.

To further contextualize our findings, it is important to consider how different exercise paradigms may influence recovery mechanisms[38]. Voluntary and forced exercise paradigms can differentially influence post-stroke recovery through both

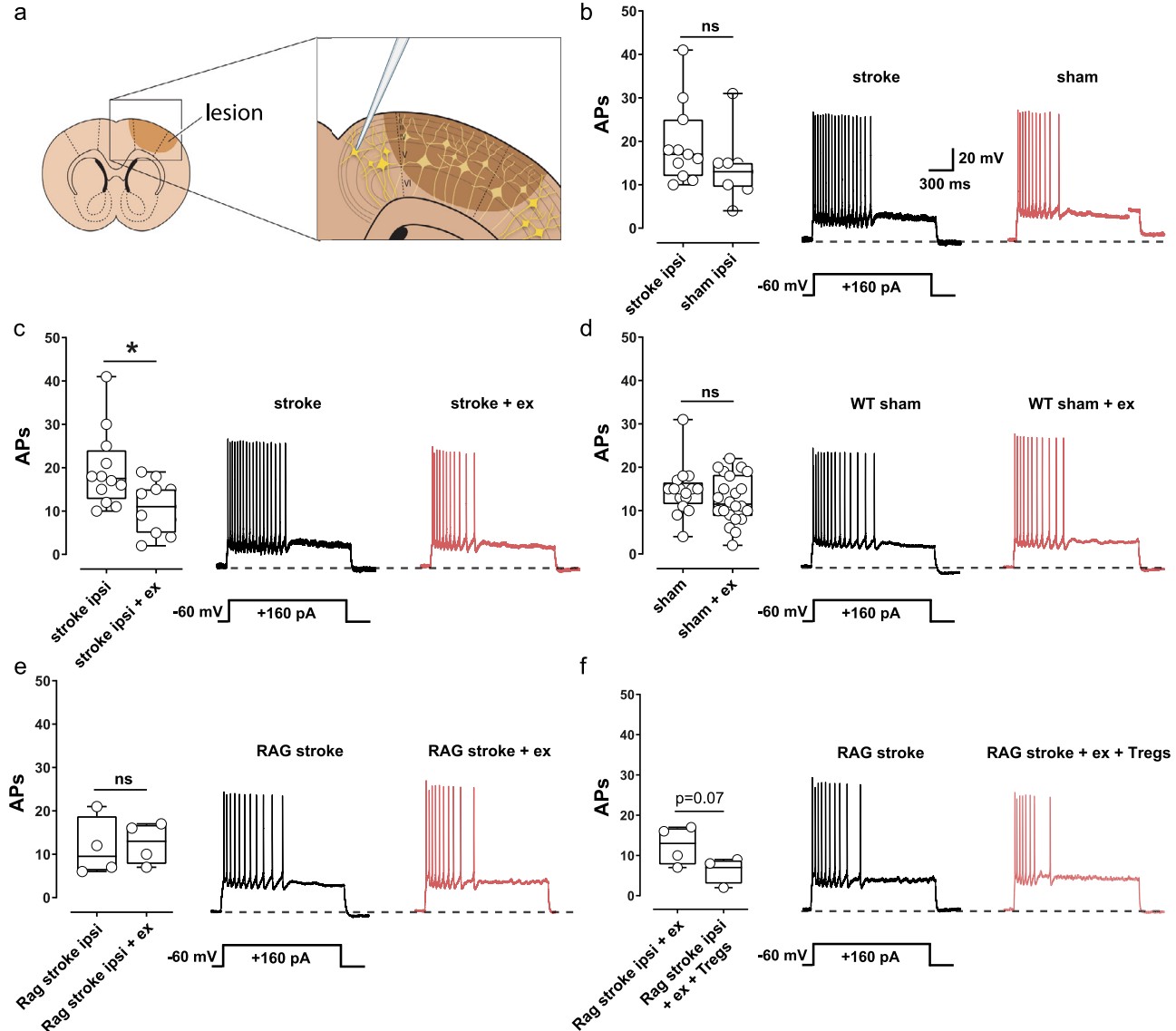

**Fig. 6 | Exercise modulates neuronal activity Treg-dependently. a** Graphic representation of a coronal slice and close-up of the periinfarct area indicating the recording site. Single-cell excitability was measured from a holding potential of -60 mV in response to a depolarizing current step (+160 pA, 2.5 s duration). **b** Fourteen days after stroke, neurons located in the periinfarct area of wild type mice showed an increased excitability as compared to sham-operated controls (*$p = 0.05$, Mann-Whitney test, $n = 9$ and 12 animals per group). **c** Exercise normalized stroke-induced hyperexcitability of periinfarct neurons in wild type mice (*$p = 0.013$, two-sided $t$-test, $n = 11$ and 12 animals per group). **d** Conversely, no exercise-mediated changes in AP generation were detected in sham animals ($p = 0.3$, Mann-Whitney test, $n = 16$

and 22 animals per group). **e** Similarly, exercise did not modulate the excitability of periinfarct neurons in RAG$^{-/-}$ mice ($p = 0.8$, two-sided $t$-test, $n = 4$ animals per group). **f** Following Treg-transfer, exercise promoted a reduced excitability of periinfarct neurons in RAG$^{-/-}$ mice, thereby suggesting that exercise modulates neuronal activity Treg-dependently ($p = 0.07$, two-sided $t$-test, $n = 4$ animals per group). For box and whisker plots, the box extends from the 25th to the 75th percentile, the center is the median and whiskers extend from the minimum or maximum. Each dot represents an individual biological replicate. Asterisks indicate levels of statistical significance: *$p < 0.05$. Source data are provided as a Source Data file.

neurotrophic and neuroimmune mechanisms. Voluntary wheel running tends to evoke robust neuroplastic benefits – for example, higher brain BDNF levels and faster motor recovery – while provoking minimal physiological stress[38]. In contrast, forced treadmill training imposes an initial stress response, evidenced by transient corticosterone elevation and anxiety-related behaviors[39]. This stress can modulate the immune milieu: acute stress triggers neuroendocrine pathways that boost immuno-suppressive cytokines like IL-10[40], potentially enhancing anti-inflammatory Treg activity. Indeed, psychological stress has been shown to augment IL-10 production via β-adrenergic signaling[40]. If unabated, however, excessive corticosterone can impair recovery—e.g., pre-stroke forced running without acclimation

exacerbated neuroinflammation and tissue damage in mice relative to non-exercised controls[39]. Crucially, animals habituate to repeated forced exercise over multi-week protocols, attenuating the stress response. Incorporating a dedicated habituation phase, as included in our experimental design, helps mitigate early stress. Notably, gradually ramping up training intensity after stroke yields better functional outcomes with lower corticosterone levels compared to abrupt high-intensity exercise[41]. Thus, with proper habituation, forced exercise can harness beneficial immunomodulation without sustained stress detriments. Importantly, the efficacy of exercise-induced immune modulation is shaped by biological variables such as age and sex. In our study, the observed benefit of exercise was critically dependent on IL-10

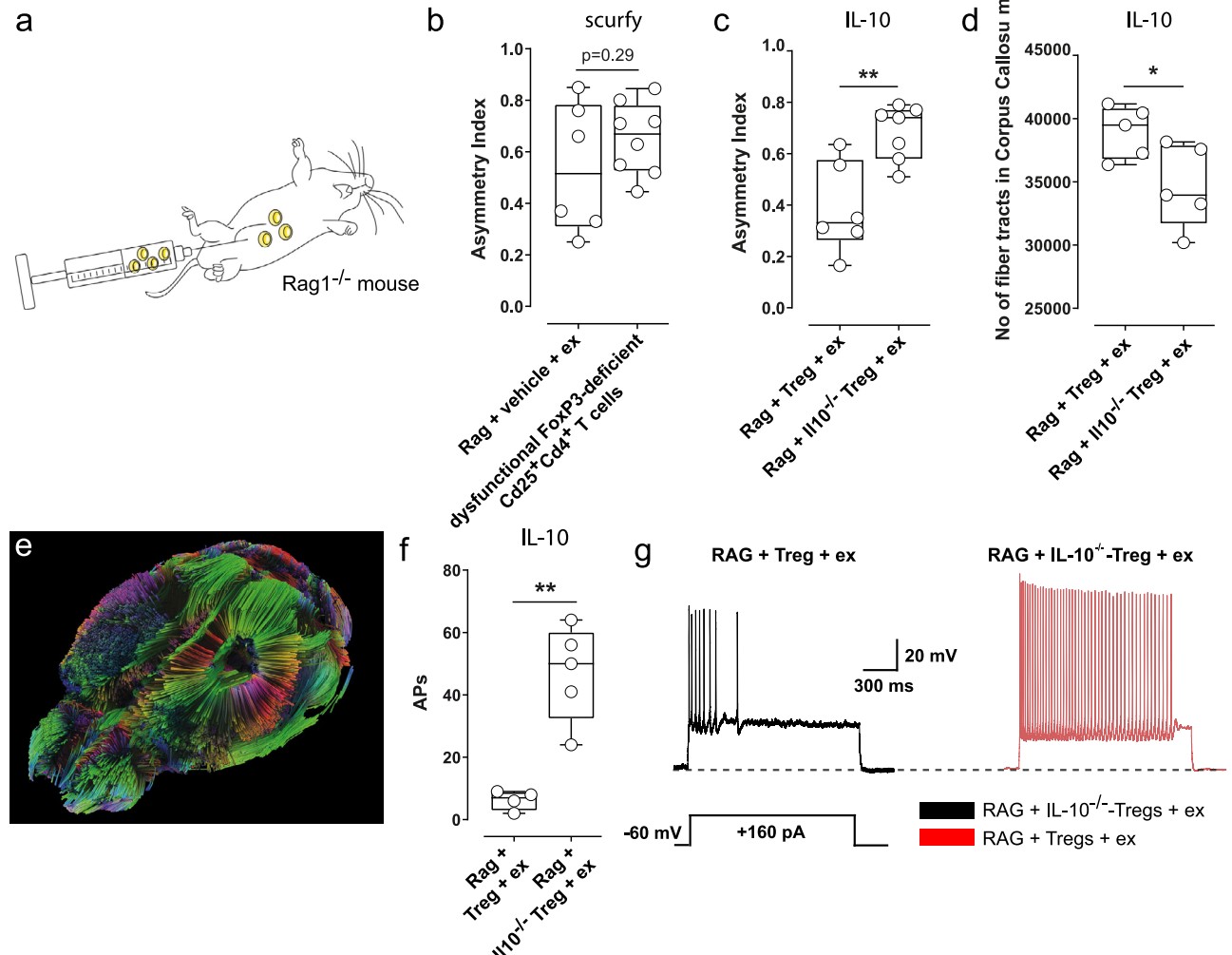

**Fig. 7 | IL-10 is a key mediator of exercise-induced stroke recovery. a** To exclude the possibility that the mere presence of activated CD4+ cells as a cofactor is sufficient to enable exercise-enhanced stroke recovery, we transferred either dysfunctional, FoxP3-deficient CD25+CD4+ T cells from scurfy mice or vehicle to RAG1−/− mice. **b** The transfer of dysfunctional CD25+CD4+ T cells did not facilitate a better functional recovery compared to vehicle ($p = 0.29$, two-sided $t$-test, $n = 8$ and 6 animals per group), thus demonstrating that the mere presence of activated CD4+ T cells is not sufficient to enable exercise-induced stroke recovery. **c** The transfer of IL-10−/−-Tregs to RAG1−/− mice resulted in significantly worse functional recovery compared to the transfer of Tregs (**$p = 0.003$, two-sided $t$-test, $n = 6$ and 7 animals per group). **d, e** Similarly, MRI-DTI fiber tracking showed a lower number of interhemispheric connections in mice that received IL-10−/−-Tregs compared to mice that received Tregs (*$p = 0.03$, two-sided $t$-test, $n = 5$ animals per group), thus

emphasizing the key role of IL-10-secretion by Tregs for neuronal network reorganization. **f, g** To clarify, if IL-10 is also crucial for Treg-effects on neuronal activity, we transferred either IL-10−/−-Tregs or Tregs to RAG1−/−-mice and assessed exercise-effects on neuronal activity of periinfarct neurons. Our results show that Treg, but not IL-10−/−-Treg, promoted exercise-mediated normalization of neuronal hyper-excitability, thus indicating that IL-10 secretion by Tregs has a pivotal role for exercise-induced neuronal repair (**$p = 0.0013$, two-sided $t$-test, $n = 4$ and 5 animals per group). For box and whisker plots, the box extends from the 25th to the 75th percentile, the center is the median and whiskers extend from the minimum or maximum. Each dot represents an individual biological replicate. Asterisks indicate levels of statistical significance: *$p < 0.05$, **$p < 0.01$ Source data are provided as a Source Data file.

signaling by Tregs, supporting the notion that immune system integrity is a prerequisite for post-stroke rehabilitation to be fully effective. Notably, all experiments were conducted in young adult male mice. Given that aging is associated with heightened post-stroke inflammation and reduced IL-10 responses[42], and that female animals exhibit distinct immunological profiles and an enhanced corticosterone response to stress[43], our findings may not fully extrapolate to aged or female populations. Future experiments will be required to systematically explore how age- and sex-related factors shape the interaction between exercise, immune modulation, and stroke recovery.

Our study has basic science and clinical translational significance. In the past, it has been shown, on the one hand, that Tregs are significant for stroke recovery and, on the other hand, that exercise improves stroke regeneration. However, the importance of Tregs for

exercise-enhanced stroke recovery had not been investigated, so far. Previous studies have highlighted crucial importance of the immune system in stroke regeneration. However, these studies focused on spontaneous recovery, whereas in clinical practice, almost all stroke patients receive exercise therapy. Our data demonstrate that an intact immune system is a prerequisite for physiotherapy to reach full efficacy and elucidate mechanisms that can be therapeutically utilized in the future, e.g., in cell-based or pharmacological treatments targeting Treg-differentiation[44].

We show that exercise modulates neuronal excitability. Although previous studies have shown that excitability changes precede neuroplasticity[14,15,25], no easily-translatable intervention that can positively influence excitability had been described so far.

In conclusion, our study demonstrates that IL-10-signaling by Tregs is a prerequisite for exercise-enhanced stroke recovery.

Mechanistically, IL-10 released by Tregs modulates the excitability of periinfarct neurons, thus increasing neuroplasticity. Particularly relevant for our daily clinical routine is the finding that physiotherapy can only be fully effective when the immune system is intact.

## Methods

### Ethical regulations

All animal procedures were performed in accordance with local animal welfare regulations and experimental protocols were approved by the local governmental authorities (Landesamt für Natur, Umwelt und Verbraucherschutz, NRW, Germany) under the approval reference number (AZ 84-02.04.2015.A476).

### Animals

We used solely adult and male mice, in detail C57BL/6 mice, RAG-1-deficient C57BL/6 mice, FoxP3-mRFP C57BL/6 mice (C57BL/6-Foxp3tm1Flv/J), scurfy mice on a C57BL/6 background (B6.Cg-Foxp3sf/Y) and IL-10-deficient C57BL/6 mice (C57BL/6-IL10tm1Cgn/J). All mice were 11–13 weeks of age, except for scurfy mice (2–3 weeks of age) and IL-10-deficient C57BL/6 mice (8–12 weeks of age). Mice were kept under standard housing conditions with a 12:12 h light-dark cycle, an ambient temperature between 20–24 °C and humidity between 45–65%. Mice had free access to pelleted food and water.

Experiments were performed in accordance with the ARRIVE guidelines (the checklist is provided in the supplement).

### Sample size calculation

A priori sample size calculations were performed to achieve 80% power to detect a relevant treatment effect of 25% with an alpha level of 0.05. We used the sample size calculator available at http://www.stat.ubc.ca.

### Photothrombotic stroke

Photothrombotic stroke was induced under general anesthesia with ketamine hydrochloride (100 mg/kg) and xylazine (10 mg/kg). The body temperature was maintained constant at 37 °C ± 0.5 °C throughout the procedure. After skull exposure, a laser beam of 4 mm diameter (Cobolt Jive ™ 75 Laser, 561 nm, Cobolt AB Sweden) was positioned 2 mm to the right of the bregma. Following an intraperitoneal injection of 0.1 ml Bengal Rose (10 mg/ml) the skull was illuminated for 20 min. In total, stroke was induced in 316 animals. Of these, 16 mice died—8 during surgery and 8 in the postoperative period. The overall mortality rate was 5.06%. Additionally, a total of 69 sham-operated mice were included in the experiments. None of the sham-operated animals died. Sham operations were performed using the same procedure, except the laser was not activated during the illumination period.

### Exercise-therapy in motorized running wheels

The animals are introduced to the treadmill 48 h after stroke induction. Initially, a 5-day habituation phase is implemented, during which the animals are placed in the treadmill for 37 min daily. Following the habituation phase, the actual training begins, with a duration of 74 min each weekday. Within these 74 min, the animals undergo three cycles in which the running speed is gradually increased from 2 m/min to 6 m/min. Behavioral tests are conducted sequentially on a single day. Food is provided ad libitum through the cage grid, with no additional interventions involved.

### Adhesive tape removal test

For bilateral sensory stimulation, two adhesive dots (25 mm²) were stuck to both palmar forepaws, and the time the animals needed to remove the adhesive dots was documented for each forepaw in three trials per day by a blinded investigator. An asymmetry score was calculated as follows: (time to remove ipsilateral dot—time to remove contralateral dot)/(time to remove ipsilateral dot + time to remove contralateral dot).

### MRI

For whole skull ex vivo diffusion-weighted magnetic resonance imaging (dMRI), skulls were fixed in 4% para-formaldehyde (PFA) for 3 days and then transferred to 1% PFA with 2 mM Magnevist (Gd-DTPA, Bayer, Leverkusen, Germany) for another 3 days with a daily change of medium. Whole skulls were embedded in 1% low-melting agarose (Sigma Aldrich, St. Louis, USA) with 2 mM Magnevist. MR images were acquired on a 9.4T horizontal animal scanner with 20 cm bore (Bruker 94/20 USR BioSpec, Avance III, Bruker BioSpin, Ettlingen, Germany), equipped with a 700 mT/m gradient system (BGA12S, BrukerBioSpin) and a 2-element cryogenic transceiver coil (Bruker BioSpin) for image acquisition. The system is operated by ParaVision 6.0.1 (Bruker BioSpin). Images were acquired with a multishot 3D spin-echo EPI sequence with the following scan parameters: echo time = 24.5 ms, repetition time = 300 ms, 8 segments, bandwidth = 250 kHz, $100 \times 100 \times 100 \, \mu m^3$ resolution, gradient duration = 4.5 ms, gradient separation = 11 ms, 80 diffusion directions, $b = 1500$ and $3000$ s/mm². Fitting of diffusion metrics and fiber tracking were performed using DSI Studio (http://dsi-studio.labsolver.org/). The restricted diffusion was quantified using restricted diffusion imaging[45]. The diffusion data were reconstructed using generalized q-sampling imaging[46] with a diffusion sampling length ratio of 1.4. Lesion volumetry was performed on the dMRI scans from a manual outline of cortical voxels with signal voids or aberrant anisotropy vectors. To quantify interhemispheric connectivity, we used a deterministic fiber tracking algorithm as previously described[47]. The midline of the corpus callosum was chosen as seed region (delineated by dominant left-right diffusion anisotropy, 50,000 seeds). The anisotropy threshold was 0.03, angular threshold 60°. The step size was 0.03 mm. Tracks with length shorter than 1 or longer than 20 mm were discarded.

### Application of an antibody specific to very late antigen-4

To prevent leukocyte immigration into the brain and thus distinguish between central and peripheral immune cell effects, we injected an antibody against very late antigen-4 (VLA-4) twice weekly. We injected 300 μg of VLA-4-specific monoclonal antibody (anti-VLA clone PS/2, isotype rat IgG2b clone LTF-2) dissolved in saline intraperitoneally (i.p.). The first injection was 24 h after ischemia-induction. Control animals received rat IgG2b isotype control monoclonal antibody. Antibodies were commercially purchased from BioXCell (Lebanon, NH, USA).

### Application of axonal tracers

Anterograde tracer injections were performed 3 weeks after photothrombotic stroke induction. Two distinct tracers were used: CB-conjugated dextran amine (CB) and biotinylated dextran amine (BDA), which were injected into the contralesional (CB) and ipsilesional (BDA) motor cortices, respectively. To this end, animals were anesthetized with ketamine hydrochloride (100 mg/kg) and xylazine (10 mg/kg) and the body temperature was maintained constant at 37 °C ± 0.5 °C. After skull exposure, burr holes were drilled +1.7 mm anteroposterior and 1.75 mm lateral to bregma (Paxinos and Franklin 2001). Injections were made through the intact dura mater with a 25-gauge, 10-μl syringe (Hamilton, Reno, NV). Deposits of 10% BDA or 10% CB (both 10,000 molecular weights; Invitrogen, Carlsbad, CA), diluted in 1.5 μl phosphate-buffered saline (PBS) were placed in 0.7 mm depth into the motor cortex. The needle remained in place for 2 min after each injection. On postlesion day 49, the animals were terminally anesthetized and perfused intracardially with 0.9% NaCl and 4% PFA.

## Tissue collection and processing for histology

Fourteen and forty-nine days after photothrombotic stroke, mice were perfused through the left ventricle with PBS for 5 min followed by 4% paraformaldehyde solution for 10 min under deep xylazine/ketamine anesthesia. Brains were removed, fixed in 4% paraformaldehyde overnight, immersed in 20% sucrose for 3 days, frozen, and stored at −80 °C.

## Immunohistochemistry

Mounted coronal mouse cryosections were rinsed three times in PBS (sections selected for amplification were additionally treated with 3% $H_2O_2$/Methanol for 10 min to block endogenous peroxidases) and thereafter incubated in Blocking Reagent (Roche Diagnostics) for 15 min to prevent nonspecific protein binding. We used the following primary antibodies for murine sections: Rabbit-anti-Arginase-1 (1:100, Abcam, ab91279), Hamster-anti-CD3 (1:50, BD Bioscience, 550277), Goat-anti-CD206 (1:100, R&D Systems, AF2535), Rat-anti-F4/80 (1:500, clone CI:A3-1, Serotec, MCA497G), Mouse-anti-GFAP (1:500, clone G-A-5, Millipore, G3893), Goat-anti-Iba1 (1:50, Abcam, ab5076), Rat-anti-Ly-6B.2 (1:100, clone 7/4, BioRad, MCA771G), Chicken-anti-MAP-2 (1:100, Abcam, ab5392), Rabbit-anti-NeuN (1:150, clone 27-4, Millipore, MABN140). Afterwards, brain slices were incubated with the appropriate Alexa Fluor secondary antibody for antigen visualization (1:100, 45 min, room temperature). Cellular nuclei were counterstained using a fluorescent preserving mounting medium containing 4′,6-diamidino-2-phenylindole (DAPI, Life, 00-4952-52). To amplify the signal of F4/80, we applied HRP-conjugated streptavidin (DAKO, Denmark, 1:100, 45 min) and biotinyl tyramide (1:100, 15 min), after incubation with respective biotinylated secondary antibodies (biotinylated anti-rat antibody, 1:100). Thereafter, amplified antigens were visualized with streptavidin-conjugated dye (Alexa Fluor594, Molecular Probes, 1:100, 45 min).

Cell quantification was performed using digitized immuno-fluorescence images within predefined areas (C1-C4, I1-I4; TC, TI, ICC1-ICC2, ICI1-ICI2) of the ipsilateral and contralateral hemispheres (see Fig. 5a). Images were acquired using a Nikon Eclipse 80i fluorescence microscope and a Zeiss AxioVision apotome (Carl Zeiss). ImageJ software 1.48 v was used for manual cell counting. The number of cells counted within the defined areas was averaged for each individual brain. Heat maps of immune cell infiltration were manually generated individually for each stained brain section and subsequently stacked and color-coded using Adobe Illustrator (Adobe Illustrator CS5).

## Quantification of crossing fibers

To evaluate midline-crossing axonal fibers, we adapted a previously published protocol[18,19]. Fibers labeled with either BDA or CB were considered to have crossed if they extended across or were located beyond an imaginary line aligned with the pyramidal tract and running parallel to the medial longitudinal fissure. This analysis was performed specifically at the level of the facial nucleus (Supplementary Fig. 3). Quantitative data are presented as the number of tracer-positive fibers per section of interest.

## Quantification of astrocytes, microglia/macrophages and T cells

Quantification of astrocytes, microglia/macrophages and T cells was performed using digitised immunofluorescence images within pre-defined areas (C1-C4, I1-I4; TC, TI, ICC1-ICC2, ICI1-ICI2) of the ipsilateral and contralateral hemispheres (see Fig. 5a). Cell count of each image was converted from cells/ROI to cells/mm². Images were acquired using a Nikon Eclipse 80i fluorescence microscope. One ROI-image covered 26.28 mm² on a 1.92 Mpixel image. ImageJ software 1.48 v was used for manual cell counting. Number of cells counted within the defined areas was averaged for each brain, individually. Heat maps of immune cell infiltration were manually generated for each stained brain section and subsequently digitized, stacked and color coded

using Adobe Illustrator (Adobe Illustrator CS5). Either increasing density (T cells) or a shift from green to yellow to red (astrocytes, microglia/macrophages) indicates an accumulation of the respective cell population within the investigated area Since T cells were found in smaller numbers when compared to glial and hematogenous cells, T cells were displayed individually as single red dot and superimposed and stacked for each condition. For the purpose of z-stack imaging a Zeiss AxioVision apotome (Carl Zeiss) was used. Z-stack step distance was ~0.4 μm.

## Blinded assessment

Functional testing, MRI measurements, quantification of histological findings and analyses of neuronal activity were performed in a blinded manner.

## Flow cytometry

Brains (ipsilateral and contralateral hemispheres without the olfactory bulb and cerebellum) were dissected after cardiac perfusion with 20 ml PBS, digested with PBS/Collagenase for 30 min at 37 °C, passed through a cell strainer and washed with PBS/1% FCS. Peripheral immune cells were obtained from cerebral cell suspensions after density centrifugation for 20 min at 2800 × $g$ in 37% Percoll without brake. Cells were washed and resuspended in PBS/2 mM EDTA/2% FCS. To obtain cervical lymph node (cLN) and splenic single cell suspensions, cLN and spleens were passed through cell strainers together with red blood cell lysis (ACK) buffer, centrifuged and resuspended in PBS supplemented with 2% FCS and 2 mM EDTA. We used the following fluorochrome-labeled antibodies: CD45 (30-11F) BV510 or FITC 1:100, CD45R/B220 (RA3-6B2) PerCP-Cy5.5 1:100, CD3 (17A2) PE-Cy7 1:200, F4/80 (BM8) APC 1:200, Ly-6G/Ly-6C (RB6-8C5) BV421 1:200, CD11c (N418) AF700 1:150, CD11b (M1/70) PE 1:800 all obtained from Biolegend and NK-1.1 and (PK136) APC-Vio770 1:200 obtained from Miltenyi Biotec, and CD3 (145-2C11) 1:200, CD4-Pacific Blue (RM4-4) 1:150, CD8a-FITC (53-6.7) 1:150, CD25-APC (PC61) 1:150, Ki-67 AF647 (B56) 1:200 from BD Biosciences, and FoxP3 (FJK-16s) PE 1:150 from eBioscience. Dead cells were identified by staining with the Fixable Viability Dye eFluor 780 (eBioscience; Thermo Fisher Scientific). After staining, cells were washed twice and resuspended in PBS with 2% FCS. Cells were acquired on a Gallios flow cytometer (Beckman Coulter) or FASC Symphony (BD Biosciences) or sorted on a FACS Aria III (BD Biosciences). Sorting was performed using an 85 μm nozzle and 4-way purity sort precision mode. Data were analyzed using DIVA or FlowJo software v10.6.1 (BD Biosciences). Cell concentrations from all tissues were manually counted in a Fuchs-Rosenthal counting chamber. Flow cytometry gating strategies are provided in Supplementary Fig. 6.

Flow cytometry was performed 14 days after stroke, as this time point reflects the full effect of the training intervention, which was initiated on day 2 and gradually intensified until day 7. Earlier time points would have likely captured only transient immune changes. To further substantiate the relevance of this time point, we conducted an additional time-course experiment using FoxP3-mRFP C57BL/6 mice (C57BL/6-Foxp3tm1Flv/J) and demonstrated that Tregs infiltrate the infarcted brain within the first 72 h after stroke, with relatively stable levels throughout the first week (see Supplementary Fig. 1).

## Isolation of T cell subsets for adoptive transfer experiments

The mouse spleen was mechanically disrupted using a 40 μm cell strainer and then rinsed with 10 ml of a washing solution containing DMEM, 1% FCS, and 1% penicillin/streptomycin. To remove erythrocytes from the splenocyte suspension, an ACK buffer (150 mM NH4Cl, 10 mM KHCO3, 0.1 mM EDTA, pH 7.3) was applied for 30 seconds, and the reaction was stopped by adding the washing medium. The resulting single cell suspensions were washed again and then resuspended in MACS buffer (PBS, 0.5% BSA, 2 mM EDTA) in preparation for the subsequent isolation of T cells subsets. To separate

CD3+-, CD8+-, CD4+- and CD4+CD25+-T cells from the spleen or lymph node, we utilized the respective T cell isolation kit (MACS, Miltenyi Biotec) following the manufacturer's instructions. In short, we incubated single cell suspensions with the respective T cell biotin-antibody mixture for 5 min at 4 °C, then incubated them with anti-biotin MicroBeads for 10 min at 4 °C. To isolate CD4+CD25+-T cells, CD25+-cells were labeled with CD25+-PE and anti-PE MicroBeads. The labeled cells were then separated from the unlabeled cell populations using magnetic field separation. To isolate FoxP3+-T cells from FoxP3-RFP reporter mice splenocytes were stained for CD4, CD8a and CD25. Regulatory T cells (CD4+CD8a-CD25+RFP+) were sorted in 4-way purity mode on the BD FACS AriaIII and collected in MACS buffer for further applications.

### RT2 profiler real time polymerase chain reaction (PCR) arrays

For RT2 profiler real time PCR, whole brain lysate was prepared 14 days after stroke. RNA was isolated using the RNeasyVR Micro Kit (Qiagen) and cDNA synthesis was performed from 500 ng total RNA using RT2 FirstStrand Kit (Qiagen). The RT2 Profiler PCR Arrays *Mouse Innate & Adaptive Immune Responses* (Cat. No. 330231 PAMM-052) and *Mouse Growth Factors* (Cat. No. 330231 PAMM-041) were conducted according to the manufacturer's protocol.

### FoxP3 expression analysis by flow cytometry

Splenic single cell suspensions were incubated with anti-CD4 PB (clone RM4-5), and anti-CD25 APC (clone, PC61 both BD Biosciences) and to exclude dead cells with the Viable Fixable Dye (FVD) 780 (eBioscience, Thermo Fisher Scientific) for 10 min. at 4 °C. After washing with PBS, intracellular FoxP3 staining was performed with the FoxP3 staining kit and the anti-FoxP3 PE antibody (clone FJK-16s, both eBioscience, Thermo Fisher Scientific) according to the manufacturer's protocol. Flow cytometric analyses were performed with LSRII using DIVA software (BD Biosciences).

### Inhibition assay

For isolation of CD4+CD25+ regulatory T cells, CD4+ T cells were enriched from single cell suspension by using the CD4+ T cell isolation kit (Miltenyi Biotech) according to the manufacturer's recommendation. After incubation with anti-CD4 PECy (clone RM4-5, Biolegend) and anti-CD25 FITC (clone 7D4, BD Biosciences) antibodies for 10 min at 4 °C, CD4+CD25+ T cells were cell-sorted with a purity of >90% by using an ARIA III Cell Sorter (BD Biosciences). CD45.1+CD4+CD25- responder T cells were isolated from splenocytes of JAXBoys mice (kindly provided by Prof. Karl Lang, University Duisburg-Essen) by using the CD4+ T cell isolation kit (Miltenyi Biotech) enclosing 0.5 μg biotinylated anti-CD25 antibody (7D4, BD Biosciences) to the antibody cocktail followed by the recommended protocol of the manufacturers. Cells were stained with the Cell Proliferation Dye eFluor670 (Thermo Fisher) according to the manufacturer's instructions. $1 \times 10^5$ CD45.1+CD4+CD25- responder T cells were either cultured alone or together with $1 \times 10^5$ sorted CD4+CD25+ regulatory T cells in 200 μl IMDM plus 10% FCS in 96 well plates in triplicate and stimulated with 1 μg/ml anti-CD3 (clone 145-2C11, BD Biosciences) in the presence of $3 \times 10^5$ irradiated splenocytes from C57BL/6 WT mice. After 72 h incubation at 37 °C, 5% $CO_2$ cells were incubated with anti-CD4 PB (clone RM4-5, BD Biosciences), and anti-CD45.1 PE (clone, A20, eBioscience, Thermo Fisher Scientific) antibodies and to exclude dead cells with the Viable Fixable Dye (FVD) 780 (eBioscience, Thermo Fisher Scientific) for 10 min. at 4 °C. T cell proliferation was assessed as loss of the proliferation dye on gated CD45.1+CD4+ responder T cells by flow cytometry using a LSRII and DIVA software (BD Biosciences).

### Analyses of neuronal activity

In compliance with effective German legal standards, animals were subjected to anesthesia-free euthanasia using DecapiCones (Braintree Scientific Inc., Braintree, MA 02185, USA). Post the rapid removal of brain tissue from the skull, brain slices (250 μm) were meticulously prepared as coronal sections in an ice-cold oxygenated solution, containing (in mM): sucrose, 200; PIPES, 20; KCl, 2.5; NaH2PO4, 1.25; MgSO4, 10; CaCl2, 0.5; dextrose, 10; with pH 7.35, adjusted using NaOH. Before initiating electrophysiological recordings, the brain slices were transferred to a chamber filled with artificial cerebrospinal fluid (ACSF), composed of (in mM): NaCl, 120; KCl, 2.5; NaH2PO4, 1.25; NaHCO3, 22; MgSO4, 2; CaCl2, 2; glucose, 25. These slices were maintained at 32 °C for 30 min. Subsequently, the slices were acclimated to room temperature (RT) before the recordings commenced. The pH of the ACSF was adjusted to 7.35 by introducing carbogen gas (95% O2 and 5% CO2).

The recording process involved patching the soma of pyramidal cortical layer IV/V neurons using an EPC-10 amplifier (HEKA Elektronik, Lamprecht, Germany). Throughout the experiment, the access resistance was monitored and ranged from 5-25 MΩ. Series resistance compensation of more than 30% was consistently applied and the software PatchMaster (HEKA Elektronik, Lamprecht, Germany) was used to control the experiment. To analyze the action potential (AP) firing pattern of cortical neurons, current-clamp recordings were performed. A +160 pA depolarizing current was applied at −60 mV for a duration of 2.5 s. FitMaster (HEKA Elektronik, Lamprecht, Germany) was utilized to quantify the number of evoked APs for comparison among conditions as well as between ipsilateral and contralateral neurons. To achieve a standardized and comparable dataset, we focused our analysis on layer IV/V excitatory neurons with consistent electrophysiological characteristics, in line with previous reports[48].

### Statistics

We used GraphPad Prism version 10 (GraphPad Software, La Jolla, CA) for statistical analyses. The data were assessed for normal distribution using the Shapiro-Wilk D'Agostino & Pearson normality test, and group comparisons were made using the student's *t*-test for normally distributed data and the Mann-Whitney test for data that were not normally distributed. Analysis of more than two groups was performed with Kruskal–Wallis test + Dunn's correction for multiple comparisons. The data are presented as mean ± SEM, and a *p* value of less than 0.05 was considered statistically significant.

### Reporting summary

Further information on research design is available in the Nature Portfolio Reporting Summary linked to this article.

## Data availability

Source data underlying Figs. 1–7 and Supplementary Figs. 1, 2, 4, and 5 are provided in the source data file. Source data are provided with this paper.

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

## Acknowledgements

This work was supported by the Deutsche Forschungsgemeinschaft (German Research Foundation, DFG) – FOR 2879 (ID 405358801, 428668629 and 507892174) and DFG Research Training Group GRK 2515. We thank Birgit Schmeddes and Sina Luppus for excellent technical assistance and Klaus Lennartz for single-cell sorting.

## Author contributions

Conception and design of the study: A.S.-P., T.R., K.D., W.H., S.G.M., J.M. Acquisition and analysis of data: A.S.-P., T.R., J.-K.S., M.H., B.M., L.W., M.C., K.D., S.L., D.S., G.M.z.H., H.A., L.A. L.H., A.M.H., L.F., L.V., P.H., V.N., C.N., K.K., J.B., E.V., T.R., C.B., E.H. Drafting manuscript and figures: A.-S.-P., J.-K.S., W.H., S.G.M., J.M. Revision and approval of manuscript: A.S.-P., T.R., J.-K.S., M.H., B.M., L.W., M.C., K.D., S.L., H.A., L.A. L.H., A.M.H., L.F., L.V., P.H., V.N., C.N., K.K., J.B., E.V., T.R., C.B., E.H., T.B., C.F., H.W., W.H., S.G.M., J.M.

## Funding

## Competing interests

The authors declare no competing interests.
