## [Transparent Peer Review file · Nature Communications]

Exercise facilitates post-stroke recovery through mitigation of neuronal hyperexcitability via Interleukin-10-signaling

Corresponding Author: Professor Jens Minnerup

Version 0:

Reviewer comments:

Reviewer #1

(Remarks to the Author)

In this study Schmidt-Pogoda et al. investigated the immune mechanisms associated with neurological recovery induced by exercise after experimental stroke. To this end, male mice underwent photothrombotic model of permanent focal cerebral ischemia, and after, the mice were forced to run in motorized running wheels. Assessment of neurological function, histological quantification of axonal plasticity and MRI-DTI fiber tracking, revealed that exercise improved functional recovery by increasing neuroplasticity, and importantly, the performance of adoptive cell transfer of different subsets of T cells, unveiled that this result was driven by a IL10-Treg cell-mediated mechanism modulating neuronal excitability in the peri-infarct.

This is an original study unveiling a novel mechanism of exercise-induced neuronal recovery after ischemic stroke. Of noteworthy is that this study describes for the first time that changes in the excitability of damaged neurons can be modulated by IL-10-release from Tregs, which makes the study relevant not only for the cerebrovascular but also for the neuroscience and neuroimmunology fields. The conclusion is reliable, and the proposed mechanism that IL-10 released by Tregs modulates the excitability of periinfarct neurons is well supported by the conducted experiments (Fig. 7). Also, the data and presentation are of relatively high quality. However, I find that the experimental designs, particularly those for the adoptive cell transfer experiments, should be described in much more detail to avoid any flaws in the interpretations of the findings. Below are my comments for considerations by the authors.

1. For all the adoptive transfers experiments, it is unknown the number of cells and when the cells were injected in mice undergoing stroke. Also, it is unknown if the same or different cell numbers of T cell types were administered in the performed experiments. Without this information, the conclusion of the study of Tregs mediating exercise-enhanced stroke recovery cannot be inferred. Treg transferring in the acute phase could induce neuroprotection affecting stroke outcome in the chronic phase, independently of a Treg direct effect impacting brain tissue remodeling (Liesz et al. 2009). Furthermore, the blocking experiments of leukocyte infiltration by anti-VLA-4 antibody treatment were initiated 24 hours after photothrombosis, which could mean that the worse functional recovery observed in this group as compared to the control group, was caused by the blocking of infiltration of immune cells in the acute phase. Furthermore, it would be helpful to perform a time course of brain infiltrating Tregs in exercised mice undergoing stroke to find out the peak of Treg migration and therefore, determine the best time point in the subacute phase to perform transferring of Tregs.
2. Of note, the authors found that exercise increased the percentage of FoxP3+CD4+ Tregs in the brain and enhanced their proliferation 14 days after stroke. Histological studies also revealed that RFP-Tre cells were anatomically localized in the infarct and peri-infarct region. However, did the authors observed that transferred Tregs were as well localized in the ischemic lesion? It would be important to demonstrate a central and not a peripheral immune cell effect of adoptively transferred Tregs.
3. The data derived from the single-cell studies of sorted CD3 cells do not add further insights to the manuscript. Indeed, after clustering analysis, a Treg cell subset was not identified, which complicates to account for the importance of a Treg role in this study. I would exclude these results from the manuscript.
4. The classification of M1/M2 macrophages should be better discussed. It is now well accepted that this dichotomous classification only reflects extremes phenotypes, but it is not representative of an in vivo scenario (Gliem et al., 2016).

5. Figure 6B. It shows $p=0.029$ instead of $p=0.29$ (line 244).

6. Please report number of excluded mice and survival rate.

7. The flow cytometry gating strategy used in the experiments and the description on how the number of cells were quantified in the histological brain sections are missing. It is also not described how heat maps of coronal brain sections were generated.

8. The obtained sequencing datasets should be submitted in a public repository such as the Gene Expression Omnibus (GEO).

References

Liesz A, Suri-Payer E, Veltkamp C, Doerr H, Sommer C, Rivest S, Giese T, Veltkamp R. Regulatory T cells are key cerebroprotective immunomodulators in acute experimental stroke. *Nat Med*. 2009 Feb;15(2):192-9. doi: 10.1038/nm.1927. Epub 2009 Jan 25. PMID: 19169263.

Gliem M, Schwaninger M, Jander S. Protective features of peripheral monocytes/macrophages in stroke. *Biochim Biophys Acta*. 2016 Mar;1862(3):329-38. doi: 10.1016/j.bbadis.2015.11.004. Epub 2015 Nov 12. PMID: 26584587.

Reviewer #2

(Remarks to the Author)

The authors investigated the impact of physical exercise using a treadmill on functional recovery, axonal sprouting, T-cell proliferation, T-cell gene expression, glial response, and neuronal excitability. The study revealed that post-stroke exercise improves functional recovery, promotes pyramidal tract and corpus callosum projections, enhances Treg proliferation, modifies the glial response, and counteracts post-stroke hyperexcitability. The authors also demonstrated that the therapeutic effects of exercise on functional recovery, corpus callosum projections, and neuronal activity depend on Treg and IL-10. Although the importance of Treg cells in stroke pathology is well-known, it is unclear if Treg cells regulate the training-induced recovery process. Thus, the finding that Treg is a prerequisite for exercise-enhanced functional and structural recovery is novel and could impact the rehabilitation research field. However, there are serious concerns about the experiment design, especially control group settings. Furthermore, their main argument that reduced hyperexcitability precedes alterations in neuronal connectivity (promotes axonal sprouting), which underlies functional improvement, contradicts previous knowledge that enhanced neuronal excitability enhances axonal sprouting and functional recovery, and the mechanism of this contradiction has not been assessed. They used standard methodology in the stroke field, but not enough detail is provided in the methods for the work to be reproduced.

The following are comments on each point.

1. The first sentence in the introduction claims that “physical training is the most effective neuroregenerative therapy in stroke patients,” but this has not been proven. The provided reference did not compare the effect of physical training with other types of therapies, such as task-specific therapy and non-invasive brain stimulation therapy (rTMS and tDCS). There are actually no scientific studies in humans that prove that exercise therapy is even effective in reproducible and large sample sizes.

2. The neuronal tracer study showed an increased axonal projection to the corticospinal tract at the level of the facial nucleus (FN) (presumably), but there is no description of why the authors chose this brain area for the observation. The quantification in the figure indicates “projections to ipsilateral FN. Did the authors quantify projections to the FN, or going by it in the corticospinal tract? The authors should explain the significance of the neuronal pathways (pyramidal tract at the facial nucleus level, corpus callosum) for functional recovery in the tape removal test. There is no indication of how this quantification was done—a example figure in a supplement would help the reader here.

3. The authors chose the tape removal test for the behavioral outcome measurement. The test is easy to perform and analyze but not very sensitive and can be affected by factors unrelated to sensorimotor function. Since the study employed various mouse lines and cell transfer methods, behavioral assessment by this test alone could mislead the conclusion. The authors should confirm their findings using multiple or more reliable tasks such as grid walk and skilled reaching tests.

4. In the patch clamp experiments, the authors concluded that exercise improves recovery and neuronal network reorganization by modulating neuronal activity Treg-dependently. However, this experiment did not assess causality between neuronal activity and recovery/network reorganization. The comparison between ipsi and contra stroke also failed to demonstrate a statistically significant difference, raising doubt about whether hyperexcitability is related to functional recovery. Moreover, hyperexcitability should be assessed in comparisons between intact and stroke brains because it is evident that the contralesional hemisphere is affected by stroke as well. Also, the response to depolarizing currents differs depending on the neuron type, cortical layer, and distance from the infarct. Thus, it is essential to provide data that demonstrate identical cell populations across groups.

5. The study did not assess the effects of treatments, such as exercise, T-cell deprivation, and cell transplantation, in intact animals. It is essential for stroke recovery study because if the treatment similarly influences intact and stroke animals, the effect is not specific to recovery. In such a way, the authors cannot conclude that exercise facilitates recovery through the Treg-IL10 mechanism.

For example, if exercise or Treg affects neuronal activity detected by patch clamp in intact animals, the decreased excitability after stroke is probably not due to suppression of hyperexcitability. Indeed, their data showed lower APs in RAG stroke contra (Fig. 6C: about 10) compared to Wild type stroke contra (Fig. 6B: about 20)

6. The T cell transfer timing is unclear in the transfer experiments. Given the time-dependent pathogenesis of stroke, this information is crucial to interpreting the data.

7. In the T cell transfer experiments in Fig. 2, the effect of different transferred cell types was compared only in stroke+exercise animals but not in non-treated stroke animals. Analysis of both non-treated and exercise animals is necessary to address whether T cells mediate exercise-induced recovery (e.g., RAG + veh, RAG + CD3, RAG + veh + ex, RAG + CD3 + ex for Fig. 2B). Similar to the issue in the patch clamp experiment (point 5), if CD3 cell transfer improves functional recovery in non-treated animals, CD3 is necessary for post-stroke recovery but not mediate exercise-induced recovery.

8. For Fig.3 data, the detailed methodology is missing. The authors must clarify the brain area of sampling (Peri-infarct? Motor cortex?), tissue dissociation methods, and cell counting methods (e.g., the definition of viable cells). Also, the reason why the authors chose to perform these analyses 14 days after the stroke should be explained. (This is also the time point for the PCR array and scRNA-seq). As described in the discussion, T cell responses can be more robust within the first week after stroke. Preferably, the authors demonstrate temporal changes of Treg and IL-10 for further analysis.

9. No description of the method for the histological analysis in Fig. 4: type of microscope, imaging methods (magnification, z-stack steps, etc.), analysis methods (heat map, calculation of cells/ROI), and software. Also, the authors must explain more details about the heatmaps (what do the color bars and red dots in the heat map represent? Are they representative or summed heatmaps?)

10. Given the PCR array results, the authors concluded that exercise induces a shift towards less inflammatory cytokine-expression profiles. However, statistical significance was detected in only a few genes from the gene panels. More definitive statistical evidence would be needed to assert general changes in inflammatory cytokine expression profiles.

11. Again, imaging and analysis methods for astrocyte/microglia/macrophage experiments are not in the manuscript.

12. As evidenced by the lack of FoxP3, the IL-10 knockout causes gene expression changes other than IL-10 and likely induces off-target effects in T cells. More specific IL-10 blocking methods are warranted to prove that the therapeutic Treg functions depend on IL-10. Similarly, the author did not show if IL-10^{-/-} Treg affects neuronal excitability in intact or non-treated stroke animals. Since the RAG+ IL-10^{-/-} Treg +ex mice showed higher APs (Fig. 7F: about 50) than non-treated stroke mice APs (Fig. 6B stroke ipsi: about 30), IL-10^{-/-} Treg likely affected the exercise-unrelated mechanism.

13. Previous studies demonstrated that post-stroke hyperexcitability or excitation by electrical/optogenetic stimulation enhances axonal sprouting. Therefore, inhibition of hyperexcitability would dampen axonal sprouting. Thus, inhibition of hyperexcitability by Treg would decrease axonal sprouting detected by neuronal tracing and MRI. Therefore, their electrophysiological data are inconsistent with neuronal connectivity data, while reduced hyperexcitability could induce better functional outcomes through different mechanisms, such as preventing excitotoxic cell death.

Reviewer #3

(Remarks to the Author)

NCOMMS-24-07563

Exercise facilitates post-stroke recovery through mitigation of neuronal hyperexcitability via Interleukin-10-signaling
This paper investigates the role of forced-use exercise on modulation of post-stroke plasticity via regulatory T cell (Treg) populations. Specifically, these experiments confirm both functional and structural plasticity changes that support recovery and are dependent on IL-10 release from Tregs post-stroke through several adoptive transfer paradigms. These are compelling, clearly presented data that include outstanding neuroimaging of fiber tracts with concomitant tract tracing, as well as electrophysiology of peri-infarct neurons showing a sustained hyperexcitability that, at least at 2 weeks, is efficaciously countered by post-stroke exercise intervention. Understanding mechanisms of lifestyle interventions such as exercise in improving long-term functional recovery are highly translationally-relevant yet rarely studied. Overall, this is a well-characterized, multifactorial series of experiments that clearly identify Treg-mediated IL-10 as an important mediator of post-stroke efficacy, with only a couple of minor concerns outlined below.

Specific concerns:

1. This paradigm is forced use exercise, which has different outcomes after brain injury than voluntary exercise. The authors should consider in the discussion both differences in exercise regimen - especially how it pertains to aspects of the immune system response re: cortisol - as well as how these effects may change with aging or in female mice. The latter is particularly relevant as both sex and age change post-stroke inflammatory responses.

2. Please include the total number of mice, as well as the total number excluded that did not exhibit post-stroke deficits or were significantly impaired.

Reviewer #4

(Remarks to the Author)

Version 1:

Reviewer comments:

Reviewer #1

(Remarks to the Author)

The authors have effectively addressed my previous comments, significantly enhancing the manuscript's clarity and depth. The revised experimental design is now more detailed, and the inclusion of additional experiments strengthens the hypothesis that regulatory T cells play a role in exercise-enhanced stroke recovery. Notably, the authors have provided data demonstrating the presence of transferred Tregs in the peri-infarct region. However, the number of transferred Tregs in this region is relatively low (<100 cells/brain), considering that 0.5 million Tregs were adoptively transferred. While this low number does not necessarily negate the potential for a localized brain effect, it raises the question of whether a peripheral effect of transferred Tregs could be modulating the systemic immune cell response, thereby affecting stroke recovery. Perhaps this point could be discussed. Additionally, while the manuscript supports an IL-10-dependent mechanism modulating the excitability of peri-infarct neurons, it does not provide evidence of a direct IL-10 effect on neurons. To strengthen this claim, experiments that specifically block or delete the IL-10 receptor in neurons should be conducted. The authors could acknowledge this limitation in the discussion section.

Reviewer #2

(Remarks to the Author)

The authors have adequately addressed all the initial queries/concerns from this review. There is one modest final concern: The authors addressed the effects of Treg on neuronal activity in additional patch clamp experiments. However, some data have small sample sizes (Supplementary Figure 4A and B: n = 4). Since three of four samples in the Sham+ex+VLA-4AB and RAG Stroke ipsi groups showed values far below the average of the Sham + ex group and Stroke ipsi, a larger sample size is necessary to conclude that the VLA-4 AB and RAG do not significantly change the number of stimulus-evoked APs.

Reviewer #3

(Remarks to the Author)

This revised paper investigates the role of forced-use exercise on modulation of post-stroke plasticity via regulatory T cell (Treg) populations. Specifically, these experiments confirm both functional and structural plasticity changes that support recovery and are dependent on IL-10 release from Tregs post-stroke through several adoptive transfer paradigms. Clearly the authors carefully considered reviewers' concerns and made great strides in amending the manuscript via text edits, additional information, as well as new experiments with control groups to further support the mechanistic hypothesis. Reporting of methods and animal inclusion/exclusions also adhere to the ARRIVE guidelines. This clarification directly supports the role of exercise modulation of the adaptive immune system to support structural and functional plasticity. Identifying mechanisms related to post-stroke efficacy, particularly with regard to a graded force exercise paradigm, is key to identifying efficacious interventions and theranostic biomarkers of lifestyle interventions to improve functional recovery for one of the leading causes of global adult-onset disability.

Reviewer #4

(Remarks to the Author)

Reviewer 1

This is an original study unveiling a novel mechanism of exercise-induced neuronal recovery after ischemic stroke. Of noteworthy is that this study describes for the first time that changes in the excitability of damaged neurons can be modulated by IL-10-release from Tregs, which makes the study relevant not only for the cerebrovascular but also for the neuroscience and neuroimmunology fields. The conclusion is reliable, and the proposed mechanism that IL-10 released by Tregs modulates the excitability of periinfarct neurons is well supported by the conducted experiments (Fig. 7). Also, the data and presentation are of relatively high quality. However, I find that the experimental designs, particularly those for the adoptive cell transfer experiments, should be described in much more detail to avoid any flaws in the interpretations of the findings. Below are my comments for considerations by the authors.

1. For all the adoptive transfers experiments, it is unknown the number of cells and when the cells were injected in mice undergoing stroke. Also, it is unknown if the same or different cell numbers of T cell types were administered in the performed experiments.

Without this information, the conclusion of the study of Tregs mediating exercise-enhanced stroke recovery cannot be inferred. Treg transferring in the acute phase could induce neuroprotection affecting stroke outcome in the chronic phase, independently of a Treg direct effect impacting brain tissue remodeling (Liesz et al. 2009). Furthermore, the blocking experiments of leukocyte infiltration by anti-VLA-4 antibody treatment were initiated 24 hours after photothrombosis, which could mean that the worse functional recovery observed in this group as compared to the control group, was caused by the blocking of infiltration of immune cells in the acute phase.

Furthermore, it would be helpful to perform a time course of brain infiltrating Tregs in exercised mice undergoing stroke to find out the peak of Treg migration and therefore, determine the best time point in the subacute phase to perform transferring of Tregs.

Ad 1, 1

We appreciate the reviewer's insightful comments regarding the adoptive transfer experiments. To clarify, in the initial experiments aiming to evaluate which T cell subtype is relevant for the observed effects, we administered 3 million total T cells per animal intraperitoneally 24 hours after ischemia induction. For Treg transfer experiments, we injected 500,000 Tregs per animal. Importantly, in all subsequent experiments, including those investigating the effects of Tregs on stroke recovery, we consistently transferred 500,000 Tregs per animal. We have included this information in the revised methods section of the manuscript for clarification.

The selection of the 24-hour time point for cell transfer was intended to target the early post-acute phase of stroke. Our rationale was to minimize the risk of missing a critical window for initiating regenerative effects, while avoiding interference with infarct development during the neuroprotective phase in this model. We acknowledge the reviewer's concern regarding the potential early neuroprotective effects of Tregs, as suggested by prior studies (Liesz et al. 2009)¹. To address this, we analyzed infarct volumes in two experimental settings: (a) wild-type mice with stroke treated with anti-VLA-4 antibody versus isotype control, and (b) RAG-deficient mice with stroke receiving Treg transfer and training versus vehicle transfer. In the first comparison, infarct volumes were $2.14 \pm 0.19 \text{ mm}^3$ in wild-type mice treated with isotype control and $1.90 \pm 0.43 \text{ mm}^3$ in wild-type mice treated with anti-VLA-4 antibody ($p = 0.60$, t-test, $n = 7$ per group, Supplementary Figure 2 A), indicating no significant difference. In the second comparison, infarct volumes were $2.03 \pm 0.27 \text{ mm}^3$ in RAG-deficient mice receiving vehicle transfer and exercise and $2.38 \pm 0.54 \text{ mm}^3$ in RAG-deficient mice receiving Treg transfer and exercise ($p = 0.54$, t-test, $n = 18$ and 6 , respectively, Supplementary Figure 2 B), again showing no significant difference. These findings suggest that the interventions improved functional recovery independently of acute neuroprotection. We have added corresponding sentences to the Results section and included the infarct volume data as Supplementary Figure 2.

Supplementary Figure 2 Infarct volume

Supplementary Figure 2: Effects of VLA-4-blockade and Treg-transfer on infarct volumes. **A** Infarct volumes of wild-type (WT) mice 24 hours after stroke treated either with isotype control antibody or anti-VLA-4 antibody. No significant differences in infarct size were observed between groups (isotype: $2.14 \pm 0.19 \text{ mm}^3$ vs. anti-VLA-4: $1.90 \pm 0.43 \text{ mm}^3$; $p = 0.60$, t-test; $n = 7$ each). **B** Infarct volumes of RAG-deficient mice 24 hours after stroke receiving either vehicle transfer and exercise or Treg transfer and exercise. Again, no significant differences in infarct size were detected (vehicle + exercise: $2.03 \pm 0.27 \text{ mm}^3$ vs. Treg transfer + exercise: $2.38 \pm 0.54 \text{ mm}^3$; $p = 0.54$, t-test; $n = 18$ and 6 , respectively).

We thank the reviewer for the suggestion to conduct a time-course analysis of brain-infiltrating Tregs in exercised mice post-stroke to determine the peak of Treg migration and have performed the following experiment:

Experiment	Animal model	Intervention	Endpoint
3	FoxP3RFP	photothrombotic stroke and running wheel exercise, starting 48 h after stroke	Heatmaps and quantification of FoxP3 cells 72 h, 96 h and 1 week after stroke

Our data show that Tregs are already present in the brain 72 hours after stroke and remain at relatively stable levels throughout the first week (see below and new Supplementary Figure 1).

Supplementary Figure 1: Effects of VLA-4 blockade and regulatory T cell transfer on infarct volume and neuroscore after stroke.

A-D Infarct volumes and neuroscores were assessed 48 hours and 72 hours after stroke induction in mice treated with an anti-VLA-4 antibody (n = 3) or isotype control (n = 2). **A** Neuroscores assessed 48 hours after stroke induction were identical in both groups (3.0 ± 0 , n = 3 and n = 2). **B** Anti-VLA-4 treatment resulted in a mean infarct volume of $23.37 \pm 1.1 \text{ mm}^3$ compared to $30.20 \pm 4.5 \text{ mm}^3$ in controls ($p = 0.15$) 48 hours after stroke. **C, D:** At 72 hours after stroke, neuroscores (2.3 ± 0.7 vs. 2.7 ± 0.3 ; $p = 0.4$) and infarct volumes ($25.34 \pm 10.76 \text{ mm}^3$ vs. $21.98 \pm 6.33 \text{ mm}^3$; $p = 0.8$, n = 3 each) were comparable between VLA-4–treated and isotype antibody-treated controls.

E-H In another cohort, infarct volume and neuroscores were measured 48 hours and 72 hours after stroke in vehicle-treated (n = 3) and Treg-transferred mice (n = 3). **E, F** At 48 hours after stroke, neuroscores (3.0 ± 0 vs. 2.0 ± 0.6 ; $p = 0.16$) and infarct volumes ($15.21 \pm 5.51 \text{ mm}^3$ vs. $17.57 \pm 7.09 \text{ mm}^3$; $p = 0.81$, n = 3 each) were comparable between both groups. **G, H** At 72 hours after stroke, neuroscores (3.0 ± 0 vs. 2.7 ± 0.3 ; $p = 0.37$) remained similar between both groups, but there was a trend towards reduced infarct volumes after Treg treatment ($10.21 \pm 2.53 \text{ mm}^3$ vs. $17.99 \pm 2.37 \text{ mm}^3$; $p = 0.05$, n = 3 per group).

2. Of note, the authors found that exercise increased the percentage of FoxP3+CD4+ Tregs in the brain and enhanced their proliferation 14 days after stroke. Histological studies also revealed that RFP-Tre cells were anatomically localized in the infarct and

peri-infarct region. However, did the authors observed that transferred Tregs were as well localized in the ischemic lesion? It would be important to demonstrate a central and not a peripheral immune cell effect of adoptively transferred Tregs.

Ad 1, 2

We thank the reviewer for raising this important point regarding the localization of adoptively transferred Tregs. To address this question, we have performed additional analyses as part of the newly included experiment, which was outlined in our response to Comment 1. Our data show that Tregs are already present in the brain 72 hours after stroke and remain at relatively stable levels throughout the first week (Supplementary Figure 1). We have included this information in the Results section under the subsection *Exercise increases the proliferation of Tregs in the ischemic brain*, where we state "In additional time-course experiments, we further demonstrated that Tregs infiltrate the infarcted brain early after stroke and are already detectable within the lesion area as early as 72 hours post-stroke, with levels remaining relatively stable throughout the first week (Supplementary Fig. 1)".

3. The data derived from the single-cell studies of sorted CD3 cells do not add further insights to the manuscript. Indeed, after clustering analysis, a Treg cell subset was not identified, which complicates to account for the importance of a Treg role in this study. I would exclude these results from the manuscript.

Ad 1,3

We thank the reviewer for this helpful comment. We agree that the single-cell RNA sequencing data did not add substantial mechanistic insight into the role of Tregs in our study and have therefore removed these results from the manuscript.

4. The classification of M1/M2 macrophages should be better discussed. It is now well accepted that this dichotomous classification only reflects extremes phenotypes, but it is not representative of an in vivo scenario (Gliem et al., 2016).

Ad 1,4

We thank the reviewer for raising this important point regarding the classification of M1/M2 macrophages. We fully agree that the traditional dichotomous classification into M1- (pro-inflammatory) and M2- (anti-inflammatory) macrophages does not fully reflect the complex and dynamic nature of macrophage polarization in vivo. As highlighted by Gliem et al. (2016)², macrophages display a broad and continuous spectrum of activation states, rather than fitting into strictly defined M1 or M2 categories.² Their phenotype is shaped by environmental cues, and they can exhibit both pro- and anti-inflammatory characteristics depending on the temporal and spatial context of the inflammatory response. Furthermore, Gliem et al. emphasize that in ischemic stroke, monocyte-derived macrophages transition from an initial inflammatory phenotype towards a more reparative state, a process that is critical for neurovascular repair and resolution of inflammation.²

In response to the reviewer's comment, we have revised the respective passage in the results section to avoid the dichotomous classification of macrophages into M1/M2 phenotypes. Instead, we now describe their functional characteristics in the context of tissue repair and inflammation resolution. In the subsection "Exercise suppresses neurotoxic astrogliosis" of the results section, where we discuss macrophage polarization after stroke, we now state:

"In the chronic stroke phase after 49 days, when microglia and macrophages are thought to contribute to tissue remodeling and regeneration² exercise led to an increased presence of these cells in the lesioned cortex (Fig. 5E, *p = 0.04, t-test), whereas their numbers were reduced in the contralateral cortex (Fig. 5E, *p = 0.04, t-test). Notably, our findings suggest that exercise promotes a shift towards a phenotype associated with tissue repair and anti-inflammatory functions²: Fourteen days after stroke, we observed an increased number of cells expressing Arginase-1, a marker commonly linked to repair-associated macrophage responses (Fig. 5 G, H, *p = 0.02, t-test). By day 49 post-stroke, a strong trend towards an

increase in CD206-expressing cells was detected, which is indicative of macrophage involvement in tissue remodeling and homeostasis (Fig. 5 G, H, $p = 0.07$, t-test). In summary, we demonstrate that exercise suppresses potentially neurotoxic astrogliosis and reduces microglial activation in the early phase of stroke recovery. Furthermore, our findings support the idea that exercise influences macrophage polarization towards a phenotype associated with repair and resolution of inflammation.^{2"}

Similarly, we have adjusted the figure legend of Figure 5 accordingly to reflect this more nuanced perspective. The revised figure legend is:

Figure 5: Exercise suppresses astrogliosis and influences macrophage polarization towards a repair-associated phenotype

A Astrocytes and microglia/macrophages were quantified in regions of interest: cortex, thalamus, and internal capsule.

B Immunohistochemical staining of glial fibrillary acidic protein (GFAP) and surface protein F4/80 was performed to visualize astrocytes and microglia/macrophages.

C, D Exercise reduced astrogliosis in the lesioned cortex and the ipsilateral thalamus 14 days post-stroke ($*p = 0.04$, t-test, and $p = 0.05$, t-test).

E, F Exercise led to decreased numbers of microglia/macrophages in the lesioned cortex and the contralateral thalamus 14 days after stroke ($*p = 0.02$, t-test, and $*p = 0.02$, Mann-Whitney test). In the chronic stroke phase after 49 days, exercise resulted in an increased number of microglia/macrophages in the lesioned cortex ($*p = 0.04$, t-test), whereas their numbers were decreased in the contralateral cortex ($*p = 0.04$, t-test).

G, H Fourteen days after stroke, we found increased numbers of cells expressing Arginase-1, a marker commonly linked to repair-associated macrophage responses ($*p = 0.02$, t-test). Forty-nine days post-stroke, we observed a rise in the quantity of cells expressing CD206, indicative of a macrophage phenotype associated with tissue remodeling and homeostasis ($*p = 0.02$, t-test).

Additionally, we have updated the discussion section to acknowledge that exercise-induced macrophage shifts are not strictly binary (M1/M2) but rather part of a more nuanced and dynamic immune response. In the revised manuscript we write: "In our experiments, the suppression of astrogliosis in exercised mice was accompanied by a reduction in microglial activation and a shift in microglia/macrophage phenotypes towards a state associated with tissue repair and inflammation resolution.² This adaptive immune response may represent an additional mechanism through which exercise promotes neuronal recovery."

We appreciate the reviewer's suggestion and believe that this revision strengthens our manuscript by aligning it with contemporary immunological concepts in stroke research.

5. Figure 6B. It shows $p=0.029$ instead of $p=0.29$ (line 244).

Ad 1,5

We sincerely thank the reviewer for pointing out this inconsistency. Unfortunately, the figure was incorrectly labeled, and we have now corrected the p-value in Figure 7 B from $p = 0.029$ to $p = 0.29$. Additionally, we have revised the group labels on the x-axis, where "RAG + Treg + ex" was mistakenly labeled and has now been corrected to "RAG + veh + ex". We appreciate the reviewer's careful attention to detail and have made the necessary adjustments accordingly.

6. Please report number of excluded mice and survival rate.

Ad 1,6

1) For the experiment investigating the effects of exercise in wild-type mice (Figure 1), we initially operated on 23 animals. Two mice died during surgery and one mouse (no ex) after one week. Additionally, two mice showed non-compliance in the adhesive tape removal test and were excluded from the analysis by a blinded examiner. Consequently, 18 animals were included in the final analysis of Figure 1 B. For the MRI examination of infarct volume (Figure 1 C) and fiber tracking (Figure 1 I), we scanned seven animals per group. The tracer injections (Figure 1 E-G) were performed in a separate cohort to avoid potential effects of stereotactic

tracer injection on functional test results and DTI fiber tracking. In this cohort, 24 animals were initially included, of which one died during surgery, and in two cases the tracer did not reach the target area (1 ex, 1 no ex).

2) For the experiment assessing the importance of T-lymphocyte (T-cell) subsets in exercise-induced stroke recovery (Figure 2), we initially operated on different groups of animals. For the experiment comparing RAG exercise versus RAG no exercise (Figure 2 B), 12 animals were operated on, with one mouse (assigned to no ex) dying postoperatively. In the experiment comparing RAG + vehicle versus RAG + CD3⁺-T cell transfer (Figure 2 C), 16 animals were operated on, with no mortality or exclusions. In the experiment comparing RAG + vehicle + exercise versus RAG + CD3⁺-T cell transfer + exercise (Figure 2 D), 25 animals were operated on, with no mortality or exclusions. For the experiment comparing RAG + CD4⁺-T cell transfer + exercise versus RAG + CD8⁺-T cell transfer + exercise (Figure 2 E), 19 animals were operated on, with one mouse dying on postoperative day 1. The varying group sizes in this experiment resulted from differences in CD4⁺- and CD8⁺-T cell yields. Similarly, for the experiment comparing RAG + FoxP3⁺-Treg + exercise versus RAG + FoxP3⁻-Treg + exercise (Figure 2 F), 17 animals were operated on, with one mouse dying on the first postoperative day. The differing group sizes were due to variations in the yield of FoxP3⁺- and FoxP3⁻-Tregs. In the experiment comparing wild-type + isotype antibody + exercise versus wild-type + anti-VLA4 antibody + exercise (Figure 2 G), 14 animals were operated on, with no mortality or exclusions. Finally, for the MRI-DTI fiber tracking experiments (Figure 2 H), 18 animals were operated on, with no mortality or exclusions. The different group sizes resulted from a limited yield of Tregs, allowing transfer in only eight animals.

In further experiments, performed to analyze the time-course of brain infiltration by Tregs in exercised mice post stroke, 9 animals were operated on, with no mortality or exclusions.

3) For the FACS analyses of brain parenchyma, cervical lymph nodes, and spleen 14 days after photothrombotic stroke (Figure 3), 23 animals were initially operated on. One mouse died on postoperative day 8, and no animals were excluded from the analysis.

4-5) For the immunohistochemical analyses on day 49 (Figures 4 and 5), the same cohort used for the ATR analysis in Figure 1b was analyzed. Initially, 23 animals were operated on, two of which died during surgery and one after one week. Consequently, 20 animals were included in the final analysis of Figure 4. For the analyses conducted after 14 days (Figures 4 and 5), a separate cohort of 12 animals was operated on, with no mortality or exclusions.

6) For the electrophysiological analyses on day 14, 24 wild-type (wt) mice underwent surgery. One of these animals (assigned to the exercise group) died on the first postoperative day. In total, 30 RAG mice were operated on, of which three died during surgery and one on the first postoperative day (two each assigned to the stroke-only and stroke + exercise groups).

Additionally, a total of 48 sham-operated wild-type mice and 21 sham-operated RAG mice were included in the experiments. None of the sham-operated animals died.

7) For the experiment comparing RAG + Treg + exercise versus RAG + dysfunctional FoxP3-deficient CD25⁺CD4⁺ T cells (Figure 7 B), 14 animals were operated on, with no mortality or exclusions. For the experiment comparing RAG + Treg + exercise versus RAG + IL-10⁻ Treg + exercise (Figure 7 C), 13 animals were operated on, with no mortality or exclusions. From this cohort, 5 animals per group were included in the MRI analyses (Figure 7 D-E).

Across all experiments, stroke was induced in 316 animals. Of these, 16 mice died – 8 during surgery and 8 in the postoperative period. The overall mortality rate was 5.06%. Additionally, a total of 69 sham-operated mice were included in the experiments. None of the sham-operated animals died. We have included this information in the revised manuscript.

7. The flow cytometry gating strategy used in the experiments and the description on how the number of cells were quantified in the histological brain sections are missing. It is also not described how heat maps of coronal brain sections were generated.

Ad 1,7

We sincerely thank the reviewer for this comment. To improve clarity and reproducibility, we have now included the flow cytometry gating strategy (Supplementary Fig. 6) and a detailed description of how cell numbers were quantified in histological brain sections.

In the methods section on immunohistochemistry, we added the following sentences "Cell quantification was performed using digitised immunofluorescence images within predefined areas (C1-C4, I1-I4; TC, TI, ICC1-ICC2, ICI1-ICI2) of the ipsilateral and contralateral hemispheres (see Figure 5a). Images were acquired using a Nikon Eclipse 80i fluorescence microscope and a Zeiss AxioVision apotome (Carl Zeiss). ImageJ software 1.48 v was used for manual cell counting. The number of cells counted within the defined areas was averaged for each individual brain. Heat maps of immune cell infiltration were manually generated individually for each stained brain section and subsequently stacked and colour coded using Adobe Illustrator (Adobe Illustrator CS5)."

8. The obtained sequencing datasets should be submitted in a public repository such as the Gene Expression Omnibus (GEO).

Ad 1,8) The sequencing datasets have been deposited in NCBI's Gene Expression Omnibus and are accessible via the GEO series accession number GSE256155. However, in line with the reviewer's recommendation, we have removed the corresponding results from the revised manuscript, as the single-cell analysis did not provide additional insights and failed to identify a distinct Treg population relevant to our study.

Reviewer 2

Reviewer #2 (Remarks to the Author):

The authors investigated the impact of physical exercise using a treadmill on functional recovery, axonal sprouting, T-cell proliferation, T-cell gene expression, glial response, and neuronal excitability. The study revealed that post-stroke exercise improves functional recovery, promotes pyramidal tract and corpus callosum projections, enhances Treg proliferation, modifies the glial response, and counteracts post-stroke hyperexcitability. The authors also demonstrated that the therapeutic effects of exercise on functional recovery, corpus callosum projections, and neuronal activity depend on Treg and IL-10. Although the importance of Treg cells in stroke pathology is well-known,

it is unclear if Treg cells regulate the training-induced recovery process. Thus, the finding that Treg is a prerequisite for exercise-enhanced functional and structural recovery is novel and could impact the rehabilitation research field. However, there are serious concerns about the experiment design, especially control group settings. Furthermore, their main argument that reduced hyperexcitability precedes alterations in neuronal connectivity (promotes axonal sprouting), which underlies functional improvement, contradicts previous knowledge that enhanced neuronal excitability enhances axonal sprouting and functional recovery, and the mechanism of this contradiction has not been assessed. They used standard methodology in the stroke field, but not enough detail is provided in the methods for the work to be reproduced.

The following are comments on each point.

1. The first sentence in the introduction claims that “physical training is the most effective neuroregenerative therapy in stroke patients,” but this has not been proven. The provided reference did not compare the effect of physical training with other types of therapies, such as task-specific therapy and non-invasive brain stimulation therapy (rTMS and tDCS). There are actually no scientific studies in humans that prove that exercise therapy is even effective in reproducible and large sample sizes.

Ad 2, 1

We thank the reviewer for this important and constructive comment. We agree that the phrasing in our original manuscript was too strong and have therefore revised the sentence to more accurately reflect the current state of evidence. The new version now reads: "Physical training is one of the most evidence-based neuroregenerative interventions in patients with stroke and has been shown to improve disability, mobility, and physical fitness, even when initiated days after symptom onset."

This revised statement reflects the available evidence while avoiding an overstatement of effectiveness compared to other therapeutic modalities.

Regarding the comment that there are no scientific studies in humans demonstrating the effectiveness of exercise therapy in reproducible and large sample sizes, we would like to refer to the Cochrane Review by Saunders et al. (2020)³, which provides a comprehensive meta-analysis of 75 randomized controlled trials involving over 3000 patients. This review concludes that cardiorespiratory and mixed training interventions significantly improve physical function and disability outcomes, and are safe and feasible components of stroke rehabilitation. While limitations such as moderate-to-low certainty of evidence in some domains and the lack of long-term follow-up data are acknowledged, the review does address concerns regarding the reproducibility and scale of the human data.

However, there are no direct comparative studies that allow for a definitive conclusion on the relative efficacy of physical training compared to other therapies such as rTMS or tDCS. We have therefore refrained from making such comparisons in our revised manuscript, acknowledging that the effectiveness of physical training is supported primarily within the context of standard rehabilitation programs, not in direct competition with other emerging neuroregenerative interventions.

2. The neuronal tracer study showed an increased axonal projection to the corticospinal tract at the level of the facial nucleus (FN) (presumably), but there is no description of why the authors chose this brain area for the observation. The quantification in the figure indicates “projections to ipsilateral FN. Did the authors quantify projections to the FN, or going by it in the corticospinal tract? The authors should explain the significance of the neuronal pathways (pyramidal tract at the facial nucleus level, corpus callosum) for functional recovery in the tape removal test. There is no indication of how this quantification was done—a example figure in a supplement would help the reader here.

Ad 2,2

We thank the reviewer for the valuable comment and constructive criticism. We appreciate the opportunity to clarify several aspects of our neuronal tracer study.

To address the question regarding our choice of anatomical localization for axonal projection analysis, we would like to provide the following explanation:

The pyramidal tract at the level of the facial nucleus is essential for functional recovery after stroke. It mediates direct voluntary motor control—including forelimb and facial movements—which is particularly relevant to performance in the adhesive tape removal (ATR) test, where forelimb coordination and sensorimotor integration are assessed. Notably, mice use their mouths to assist in removing the adhesive, engaging facial musculature and thereby emphasizing the relevance of FN-associated motor circuits. The quantification of projections to the ipsilateral FN was intended to capture structural reorganization and compensatory mechanisms following stroke. Since the exercise intervention in our experiments had a significant effect on performance in the adhesive tape removal test, this region is of particular importance. Furthermore, our approach follows established and validated methods, as described in the studies by Reitmeir et al.⁴ and Minnerup et al.⁵, which have demonstrated the relevance of FN-associated corticospinal projections in post-stroke neuroplasticity.

To justify the choice of the observation area directly within the results section—making it easier for the reader to understand the rationale in the context of our findings—we included the following paragraph into the results section: "Forty-nine days after stroke, we quantified crossing fibers at the level of the facial nucleus. This anatomical region is particularly relevant for functional recovery, as the pyramidal tract at the level of the facial nucleus mediates voluntary motor control of the forelimbs and facial musculature—both of which are engaged during the adhesive tape removal test used in our study. Notably, mice assist tape removal with their mouths, further emphasizing the functional importance of this area. The analysis at this level was therefore chosen to capture structural reorganization in a circuit directly linked to the behavioral improvements observed after training."

For the analysis, we used a previously established and validated method described by Reitmeir et al.⁴, and Minnerup et al⁵, adopted from Lindau et al⁶. We added the following detailed description of the technique to the Methods section of the revised manuscript. Additionally, we included a schematic figure in the supplement to illustrate the quantification procedure and support reproducibility.

"To evaluate midline-crossing axonal fibers, we adapted a previously published protocol^{6,7}. Fibers labeled with either BDA or CB were considered to have crossed if they extended across or were located beyond an imaginary line aligned with the pyramidal tract and running parallel to the medial longitudinal fissure. This analysis was performed specifically at the level of the facial nucleus (Supplementary Fig. 3). Quantitative data are presented as the number of tracer-positive fibers per section of interest."

3. The authors chose the tape removal test for the behavioral outcome measurement. The test is easy to perform and analyze but not very sensitive and can be affected by factors unrelated to sensorimotor function. Since the study employed various mouse lines and cell transfer methods, behavioral assessment by this test alone could mislead the conclusion. The authors should confirm their findings using multiple or more reliable tasks such as grid walk and skilled reaching tests.

Ad 2,3

We thank the reviewer for this thoughtful and important comment. In our baseline experiments, we included both the adhesive tape removal (ATR) test and the foot fault test as measures of sensorimotor performance. However, in our photothrombosis model, the foot fault test revealed only mild and transient deficits, making it less suitable for assessing long-term functional recovery. In contrast, the ATR test consistently detected significant and persistent group differences over time, providing a more robust and sensitive behavioral readout in our setting. The ATR test has been shown in the literature to be particularly sensitive following

photothrombotic stroke, with measurable effects persisting over several weeks (Bouet et al., 2009; Ruan & Yao, 2020)^{8,9}. In addition to its sensitivity, the ATR test captures not only forelimb sensorimotor function but also hand-mouth coordination. This is especially relevant in our study, as we observed initial impairments in this coordination, which improved with our intervention. Notably, this behavioral improvement aligns with the anatomical findings of axonal sprouting at the level of the facial nucleus, suggesting a meaningful relationship between structural and functional recovery. Nevertheless, we agree with the reviewer that the use of only one behavioral test represents a limitation of our study, which we have now acknowledged in the discussion. In our revised discussion, we added the following sentences "Our study is limited by the use of a single behavioral test to assess functional recovery, which may not capture all aspects of sensorimotor performance. However, the adhesive tape removal test has proven to be particularly sensitive and robust in our stroke model^{8,9} and aligns well with the anatomical and functional effects observed in our study."

4. In the patch clamp experiments, the authors concluded that exercise improves recovery and neuronal network reorganization by modulating neuronal activity Treg-dependently. However, this experiment did not assess causality between neuronal activity and recovery/network reorganization. The comparison between ipsi and contra stroke also failed to demonstrate a statistically significant difference, raising doubt about whether hyperexcitability is related to functional recovery. Moreover, hyperexcitability should be assessed in comparisons between intact and stroke brains because it is evident that the contralesional hemisphere is affected by stroke as well. Also, the response to depolarizing currents differs depending on the neuron type, cortical layer, and distance from the infarct. Thus, it is essential to provide data that demonstrate identical cell populations across groups.

Ad 2,4

We thank the reviewer for the critical and constructive comments regarding our electrophysiological data and conclusions. To address reviewer comments 4, 5, and 12, we have performed a comprehensive set of additional experiments.

As expected by the reviewer, we were not able to confirm a statistically significant difference in neuronal excitability between the ipsilateral and contralateral hemispheres after stroke. However, when comparing stroke-affected mice to sham-operated controls, we did observe a significant increase in excitability. These data confirm that stroke induces hyperexcitability (see revised Figure 6 B).

Revised Figure 6 B Cortical neurons recorded in the perilesional hemisphere of stroke animals display a significant increase in AP generation, as compared to sham controls ($p = 0.049$, t-test).

We acknowledge that the mouse motor cortex comprises a heterogeneous population of neurons with varying electrophysiological properties, depending on neuron type and location within and across cortical layers. To improve clarity regarding our experimental design, we have revised our patch-clamp dataset accordingly.

As retrospective identification of neuron types based solely on their anatomical location was not feasible, we classified recorded neurons based on their firing patterns, following the approach described by Scala et al. (2021)¹⁰. In a first step, GABAergic interneurons were identified using established electrophysiological criteria (Scala et al. 2019)¹¹ and excluded from

the analysis. As a result, our analysis focused exclusively on layer IV/V excitatory neurons exhibiting consistent and well-characterized electrophysiological properties, in line with previous reports¹⁰. This information has been added to the Methods section for clarity and transparency. Representative current-clamp traces in the revised manuscript have been updated to reflect these changes.

The neurons analyzed in the patch-clamp experiments were selected from layer IV/V pyramidal neurons based on their firing patterns. In the anterograde tracing experiments, we likewise analyzed axons from the pyramidal tract. Together, these approaches demonstrate structural remodeling and altered neuronal activity within the same neuronal population, providing a coherent mechanistic link to the observed functional improvements in the adhesive tape removal test. Although this correlation does not establish causality, it supports the functional relevance of the analyzed neurons for motor circuit reorganization and behavioral recovery. Further, our findings are consistent with existing explanatory models. Carmichael et al. have shown that *hypoexcitability* impairs recovery and that plasticity can be initiated through a transient increase in excitability¹². In contrast, our data do not suggest a reduction in excitability but rather a normalization of *hyperexcitability* induced by stroke. Persistent hyperexcitability has been associated with impaired recovery and may even contribute to secondary neurodegeneration, as reported by others¹³⁻¹⁷. Therefore, while our observations do not establish final causality, they integrate well into the current understanding of post-stroke plasticity and recovery mechanisms.

5. The study did not assess the effects of treatments, such as exercise, T-cell deprivation, and cell transplantation, in intact animals. It is essential for stroke recovery study because if the treatment similarly influences intact and stroke animals, the effect is not specific to recovery. In such a way, the authors cannot conclude that exercise facilitates recovery through the Treg-IL10 mechanism.

For example, if exercise or Treg affects neuronal activity detected by patch clamp in intact animals, the decreased excitability after stroke is probably not due to

suppression of hyperexcitability. Indeed, their data showed lower APs in RAG stroke contra (Fig. 6C: about 10) compared to Wild type stroke contra (Fig. 6B: about 20)

Ad 2,5

We thank the reviewer for highlighting this point. We performed additional patch-clamp recordings accordingly, to explore the impact of exercise on neuronal excitability in intact animals. Here, we observed that exercise did not alter the excitability of cortical neurons in sham (intact) animals, whereas a significant exercise-mediated reduction in AP generation was detected in stroke animals (revised Figure 6 B). To further investigate mechanisms influencing neuronal excitability in intact animals, we additionally assessed the role of immune cells by blocking leukocyte–endothelial interactions using in mice which were administered a VLA-4 antibody (AB). In comparison to untreated WT sham animals, neither the isotype control AB nor the VLA-4 AB induced significant changes in the number of stimulus-evoked APs (Supplementary Fig. 4 A). These findings support the notion that the modulatory effects of exercise and immune cells are specifically relevant in the context of stroke and might exert a distinct role in recovery processes. We agree with the reviewer’s remark that neurons recorded in the contralateral hemisphere of RAG mice were characterized by reduced excitability, as compared to WT stroke mice (Supplementary Fig. 4 C). Noteworthy, no significant differences were observed when the excitability of neurons located in the perilesional hemispheres were compared (Supplementary Fig. 4 B). As all experimental comparisons within our manuscript were performed within the same genotype (WT vs. WT and RAG vs. RAG), we are confident that these baseline variations do not affect the overall validity of our conclusions.

We added the following sentences to the results section of our revised manuscript: "To confirm that the immune cell-dependent effects of exercise on neuronal excitability are specific to the stroke context, we performed additional patch-clamp recordings in sham-operated wild-type animals. In these experiments, blockade of leukocyte–endothelial interactions using a VLA-4 antibody did not affect cortical neuron excitability in exercised sham animals (Supplementary

Fig. 4 A). These results further substantiate the specificity of the observed effects to the post-stroke condition. Comparison of wild-type and RAG mice revealed no significant difference in neuronal excitability on the ipsilateral side (Supplementary Fig. 4 B), but a significant genotype effect on the contralateral side (Supplementary Fig. 4 C); however, since identical genotypes were compared within each experimental group, we do not consider this finding to critically impact our conclusions."

Supplementary Figure 4: Control conditions strengthen the specific impact of exercise on altered cortical excitability in stroke.

A Blockade of leukocyte–endothelial interaction with isotype control antibody (AB), or with VLA-4 AB treatment, does not alter cortical neuron excitability in WT sham animals undergoing exercise ($p > 0.05$, Kruskal–Wallis test + Dunn’s correction, $n = 4-22$). **B, C** Comparison of AP counts between stroke WT and stroke RAG mice shows no significant difference in excitability on the ipsilateral side (**B**, $p = 0.13$, t- test, $n = 4-12$), but reveals a significant genotype effect on the contralateral side (**C**, $*p = 0.012$, t- test, $n = 5-12$). **D** IL-10^{-/-} Tregs do not further exacerbate hyperexcitability in RAG sham animals without exercise ($p = 0.32$, t- test, $n = 9-12$).

6. The T cell transfer timing is unclear in the transfer experiments. Given the time-dependent pathogenesis of stroke, this information is crucial to interpreting the data.

Ad 2,6

We thank the reviewer for this important comment. In our experiments, the T cell transfer was performed one day after induction of ischemia. We agree that this timing is critical for the interpretation of the data and included this information explicitly in the revised version of the manuscript. The selection of the 24-hour time point was based on our intention to target the post-acute phase of stroke — after the neuroprotective window in our model — in order to modulate regenerative processes without confounding effects on infarct development. While our rationale was to minimize the risk of missing a critical window for initiating regenerative responses, we acknowledge concerns regarding potential early neuroprotective effects of Tregs, as suggested by previous studies (Liesz et al., 2009)¹. To further address the reviewer's concern regarding the timing and potential neuroprotective effects of Treg transfer, we analyzed infarct volumes in the transfer and blockade experiments and present the results in Supplementary Fig. 2. Specifically, infarct volumes were comparable between wild-type mice treated with anti-VLA-4 antibody versus isotype control ($2.14 \pm 0.19 \text{ mm}^3$ vs. $1.90 \pm 0.43 \text{ mm}^3$; $p = 0.60$, t-test; $n = 7$ each) and between RAG-deficient mice receiving Treg transfer and exercise versus vehicle transfer and exercise ($2.03 \pm 0.27 \text{ mm}^3$ vs. $2.38 \pm 0.54 \text{ mm}^3$; $p = 0.54$, t-test; $n = 18$ and 6 , respectively). These findings support that the observed functional improvements were not the result of acute neuroprotective effects.

7. In the T cell transfer experiments in Fig. 2, the effect of different transferred cell types was compared only in stroke+exercise animals but not in non-treated stroke animals. Analysis of both non-treated and exercise animals is necessary to address whether T cells mediate exercise-induced recovery (e.g., RAG + veh, RAG + CD3, RAG + veh + ex, RAG + CD3 + ex for Fig. 2B). Similar to the issue in the patch clamp experiment (point 5), if CD3 cell transfer improves functional recovery in non-treated animals, CD3 is necessary for post-stroke recovery but not mediate exercise-induced recovery.

Ad 2, 7

We thank the reviewer for this important observation. In response, we have added a new panel (Fig. 2 C) comparing RAG + vehicle without exercise to RAG + CD3 without exercise. This comparison shows no difference in functional recovery, indicating that T cell transfer alone is not sufficient to promote recovery after stroke.

Figure 2 D (previously Figure 2 C) now illustrates the comparison between RAG + vehicle + exercise and RAG + CD3 + exercise. Here, we observe that exercise alone, in the absence of T cells, does not promote recovery—whereas exercise combined with T cell transfer does.

Taken together, these results demonstrate that T cells are required to mediate the beneficial effects of exercise on post-stroke recovery.

Revised Figure 2, Panels B-C. **B** Running wheel training did not influence the asymmetry index in the adhesive tape removal test ($p = 0.46$, t-test, $n = 6$ and 5) in $RAG1^{-/-}$ mice without adoptive T cell-transfer, suggesting that exercise does not improve the recovery of $RAG1^{-/-}$ mice. **C** Adoptive transfer of $CD3^{+}$ T cells alone, in the absence of exercise, was also insufficient to enhance recovery in $RAG1^{-/-}$ mice ($p = 0.28$, t-test, $n = 8$ and 8 per group). **D** Following an adoptive transfer of $CD3^{+}$ -T cells, running wheel training led to a significantly

reduced asymmetry index in the adhesive tape removal test (**p = 0.0002, t-test, n = 13 and 12 per group), which indicates improved functional outcome.

8. For Fig.3 data, the detailed methodology is missing. The authors must clarify the brain area of sampling (Peri-infarct? Motor cortex?), tissue dissociation methods, and cell counting methods (e.g., the definition of viable cells). Also, the reason why the authors chose to perform these analyses 14 days after the stroke should be explained. (This is also the time point for the PCR array and scRNA-seq). As described in the discussion, T cell responses can be more robust within the first week after stroke. Preferably, the authors demonstrate temporal changes of Treg and IL-10 for further analysis.

Ad 2,8

We thank the reviewer for pointing out the need for a more detailed description of the methodology related to the flow cytometry experiments in Figure 3.

For flow cytometric analysis, brains (ipsilateral and contralateral hemispheres, excluding the olfactory bulb and cerebellum) were collected following cardiac perfusion with PBS. Tissue was enzymatically digested using collagenase, filtered through a cell strainer, and washed with PBS containing fetal calf serum (FCS). Peripheral immune cells were isolated from the resulting cerebral cell suspensions using Percoll density gradient centrifugation. Dead cells were excluded from the analysis based on staining with Fixable Viability Dye eFluor 780. A detailed description of the procedure has now been included in the Materials and Methods section of the revised manuscript.

With regard to the selected time point (14 days post-stroke), we agree that T cell responses are particularly dynamic during the first week after stroke, as also discussed in the literature. However, in our study, running wheel training was initiated on day 2 after stroke and gradually intensified until day 7. Therefore, we chose day 14 for analysis to ensure the assessment

reflects the full effect of the training intervention. Earlier time points would likely have captured only partial or transient immune responses to exercise.

To assess the temporal dynamics of infiltrating Tregs, we have performed an additional experiment using FoxP3RFP mice subjected to photothrombotic stroke followed by voluntary running wheel exercise, starting 48 hours after stroke induction

Experiment	Animal model	Intervention	Endpoint
3	FoxP3RFP	photothrombotic stroke and running wheel exercise, starting 48 h after stroke	Heatmaps and quantification of FoxP3 cells 72 h, 96 h and 1 week after stroke

Our results show that Tregs are already present in the brain 72 hours after stroke and remain at relatively stable numbers throughout the first week. These findings support the relevance of

our previously chosen time point at day 14 to study the sustained effects of Treg presence and exercise on post-stroke immune regulation.

9. No description of the method for the histological analysis in Fig. 4: type of microscope, imaging methods (magnification, z-stack steps, etc.), analysis methods (heat map, calculation of cells/ROI), and software. Also, the authors must explain more details about the heatmaps (what do the color bars and red dots in the heat map represent? Are they representative or summed heatmaps?)

Ad 2,9

We thank the reviewer for this valuable comment. In response, we have added the following detailed description of the histological analysis methods to the revised Methods section, under the newly introduced subsection "Quantification of astrocytes, microglia/macrophages and T cells": "Quantification of astrocytes, microglia/macrophages and T cells was performed using digitized immunofluorescence images within predefined areas (C1-C4, I1-I4; TC, TI, ICC1-ICC2, ICI1-ICI2) of the ipsilateral and contralateral hemispheres (see Figure 5a). Cell count of each image was converted from cells/ROI to cells/mm². Images were acquired using a Nikon Eclipse 80i fluorescence microscope. One ROI-image covered 26.28 mm² on a 1.92 Mpixel image. ImageJ software 1.48 v was used for manual cell counting. Number of cells counted within the defined areas was averaged for each brain, individually. Heat maps of immune cell infiltration were manually generated for each stained brain section and subsequently digitized, stacked and color coded using Adobe Illustrator (Adobe Illustrator CS5). Either increasing density (T cells) or a shift from green to yellow to red (astrocytes, microglia/macrophages) indicates an accumulation of the respective cell population within the investigated area. Since T cells were found in smaller numbers when compared to glial and hematogenous cells, T cells were displayed individually as single red dot and superimposed and stacked for each condition.

For the purpose of z-stack imaging a Zeiss AxioVision apotome (Carl Zeiss) was used. Z-stack step distance was approx. 0.4 μm ."

10. Given the PCR array results, the authors concluded that exercise induces a shift towards less inflammatory cytokine-expression profiles. However, statistical significance was detected in only a few genes from the gene panels. More definitive statistical evidence would be needed to assert general changes in inflammatory cytokine expression profiles.

Ad 2,10

We agree with the reviewer's comment and have rewritten the corresponding paragraph in the results section of the revised manuscript.

11. Again, imaging and analysis methods for astrocyte/microglia/macrophage experiments are not in the manuscript.

Ad 2,11

We thank the reviewer for this helpful remark. We have addressed this point together with our response to Comment 9, where we now provide detailed information on the imaging and analysis methods used for the astrocyte, microglia, and macrophage experiments.

12. As evidenced by the lack of FoxP3, the IL-10 knockout causes gene expression changes other than IL-10 and likely induces off-target effects in T cells. More specific IL-10 blocking methods are warranted to prove that the therapeutic Treg functions depend on IL-10.

Similarly, the author did not show if IL-10^{-/-} Treg affects neuronal excitability in intact or non-treated stroke animals.

Since the RAG⁺ IL-10^{-/-} Treg +ex mice showed higher APs (Fig. 7F: about 50) than non-treated stroke mice APs (Fig. 6B stroke ipsi: about 30), IL-10^{-/-} Treg likely affected the exercise-unrelated mechanism.

Ad 2,12

We thank the reviewer for this remark, which gives us the opportunity to bring additional clarity to the study.

As suggested, we further explored the effect of IL-10^{-/-} Tregs on neuronal excitability in RAG sham animals without exercise. Here, no significant differences emerged when comparing the number of stimulus-evoked APs (Supplementary Fig. 4 D). This finding supports the central role of IL-10 in Treg-mediated recovery and exercise-modulated neuronal excitability after stroke. We added the following sentences to the results section of our revised manuscript: "To verify that IL-10-deficient Tregs do not affect neuronal excitability independently of stroke, we performed additional patch-clamp recordings in RAG sham animals receiving IL-10^{-/-} Tregs. These recordings revealed no significant changes in neuronal excitability compared to untreated sham controls (Supplementary Fig. 4 D)"

Supplementary Figure 4 D. IL-10^{-/-} Tregs do not further exacerbate hyperexcitability in RAG sham animals without exercise (p > 0.05, unpaired two-tailed Student's *t*-test).

Given the broad regulatory functions of IL-10, contributions beyond the exercise-related mechanisms cannot be entirely ruled out.

To assess whether other potential off-target effects resulting from IL-10 deficiency might influence the general inhibitory capacity of IL-10-deficient CD4⁺CD25⁺ Tregs, we performed a co-culture experiment *in vitro*. As depicted in Supplementary Fig. 5 B, CD4⁺CD25⁺ Tregs from IL-10 deficient mice exhibit the same suppressive activity in the IL-10 independent *in vitro* setting as IL-10 proficient CD4⁺CD25⁺ Tregs from WT mice. From this result, we conclude that the therapeutic function of Tregs in the *in vivo* situation indeed depends on Treg-derived IL-10.

B
Supplementary Fig. 5: IL-10 deficiency has no impact on FoxP3 expression and inhibitory activity of Tregs. **A** FoxP3 expression in gated CD4⁺CD25⁺ Tregs from IL-10 deficient (IL-10KO) and C57BL/6 (WT) mice analyzed by flow cytometry. **B** Proliferation of CD4⁺CD25⁻ responder cells (resp) stimulated with anti-CD3 and cultured alone or co-cultured with CD4⁺CD25⁺ Tregs from IL-10 deficient (IL-10KO) and C57BL/6 (WT) mice at a ratio 1:1 was analyzed by flow cytometry. Results are summarized as mean +/- SD from n = 2 – 5 mice. ****p<0.001, One-Way ANOVA.

13. Previous studies demonstrated that post-stroke hyperexcitability or excitation by electrical/optogenetic simulation enhances axonal sprouting. Therefore, inhibition of hyperexcitability would dampen axonal sprouting. Thus, inhibition of hyperexcitability by Treg would decrease axonal sprouting detected by neuronal tracing and MRI. Therefore, their electrophysiological data are inconsistent with neuronal connectivity data, while reduced hyperexcitability could induce better functional outcomes through different mechanisms, such as preventing excitotoxic cell death.

Ad 2,13

We thank the reviewer for this thoughtful comment and the opportunity to clarify the relationship between neuronal excitability, structural plasticity, and functional recovery in our study.

It was shown that *hypoexcitability* impairs recovery and that plasticity can be initiated through a transient increase in excitability¹². However, our data do not suggest a reduction in excitability below physiological levels, but rather a *normalization* of stroke-induced *hyperexcitability*, as evidenced by the comparison between stroke and sham animals (Figure 6 B). This is in line with previous reports showing that persistent hyperexcitability is detrimental for recovery, as it has been linked to impaired functional outcomes and secondary neurodegeneration¹³⁻¹⁶. From this perspective, restoring excitability to a physiological range may actually support recovery and structural remodeling by preventing excitotoxicity, rather than impairing it.

Therefore, our electrophysiological findings are not in contradiction with our neuronal connectivity data, but rather complementary. These results can be well integrated into current concepts of post-stroke neuroplasticity, emphasizing the importance of a tightly regulated balance of excitability to support recovery.

To improve clarity, we have added the following sentences to the discussion of our revised manuscript: "Our electrophysiological data indicate that exercise normalizes stroke-induced hyperexcitability rather than reducing neuronal excitability below physiological levels. This normalization may prevent excitotoxicity and support structural remodeling, aligning with current concepts of post-stroke neuroplasticity that emphasize the importance of a tightly regulated balance of excitability to enable neuroplasticity."

Reviewer #3 (Remarks to the Author):

NCOMMS-24-07563

Exercise facilitates post-stroke recovery through mitigation of neuronal

hyperexcitability via Interleukin-10-signaling

This paper investigates the role of forced-use exercise on modulation of post-stroke plasticity via regulatory T cell (Treg) populations. Specifically, these experiments confirm both functional and structural plasticity changes that support recovery and are dependent on IL-10 release from Tregs post-stroke through several adoptive transfer paradigms. These are compelling, clearly presented data that include outstanding neuroimaging of fiber tracts with concomitant tract tracing, as well as electrophysiology of peri-infarct neurons showing a sustained hyperexcitability that, at least at 2 weeks, is efficaciously countered by post-stroke exercise intervention. Understanding mechanisms of lifestyle interventions such as exercise in improving long-term functional recovery are highly translationally-relevant yet rarely studied. Overall, this is a well-characterized, multifactorial series of experiments that clearly identify Treg-mediated IL-10 as an important mediator of post-stroke efficacy, with only a couple of minor concerns outlined below.

Specific concerns:

1. This paradigm is forced use exercise, which has different outcomes after brain injury than voluntary exercise. The authors should consider in the discussion both differences in exercise regimen - especially how it pertains to aspects of the immune system response re: cortisol - as well as how these effects may change with aging or in female mice. The latter is particularly relevant as both sex and age change post-stroke inflammatory responses.

Ad 3,1

We thank the reviewer for this important comment. Indeed, the type of exercise paradigm plays a critical role in shaping the immune, hormonal, and neuroplastic response to stroke rehabilitation. In response, we have now included a dedicated paragraph in the discussion addressing the differences between forced and voluntary exercise, their respective impacts on

the immune system (particularly Treg/IL-10 responses and stress hormone signaling), and potential age- and sex-related effects. Specifically, we added the following passage:

"To further contextualize our findings, it is important to consider how different exercise paradigms may influence recovery mechanisms¹⁸. Voluntary and forced exercise paradigms can differentially influence post-stroke recovery through both neurotrophic and neuroimmune mechanisms. Voluntary wheel running tends to evoke robust neuroplastic benefits – for example, higher brain BDNF levels and faster motor recovery – while provoking minimal physiological stress¹⁸. In contrast, forced treadmill training imposes an initial stress response, evidenced by transient corticosterone elevation and anxiety-related behaviors¹⁹. This stress can modulate the immune milieu: acute stress triggers neuroendocrine pathways that boost immunosuppressive cytokines like IL-10²⁰, potentially enhancing anti-inflammatory Treg activity. Indeed, psychological stress has been shown to augment IL-10 production via β -adrenergic signaling²⁰. If unabated, however, excessive corticosterone can impair recovery – e.g., pre-stroke forced running without acclimation exacerbated neuroinflammation and tissue damage in mice relative to non-exercised control¹⁹. Crucially, animals habituate to repeated forced exercise over multi-week protocols, attenuating the stress response. Incorporating a dedicated habituation phase, as included in our experimental design, helps mitigate early stress. Notably, gradually ramping up training intensity after stroke yields better functional outcomes with lower corticosterone levels compared to abrupt high-intensity exercise²¹. Thus, with proper habituation, forced exercise can harness beneficial immunomodulation without sustained stress detriments. Importantly, the efficacy of exercise-induced immune modulation is shaped by biological variables such as age and sex. In our study, the observed benefit of exercise was critically dependent on IL-10 signaling by Tregs, supporting the notion that immune system integrity is a prerequisite for post-stroke rehabilitation to be fully effective. Notably, all experiments were conducted in young adult male mice. Given that aging is associated with heightened post-stroke inflammation and reduced IL-10 responses²², and that female animals exhibit distinct immunological profiles and an enhanced corticosterone response to stress²³, our findings may not fully extrapolate to aged

or female populations. Future studies will be required to systematically explore how age- and sex-related factors shape the interaction between exercise, immune modulation, and stroke recovery."

2. Please include the total number of mice, as well as the total number excluded that did not exhibit post-stroke deficits or were significantly impaired.

Ad 3,2

1) For the experiment investigating the effects of exercise in wild-type mice (Figure 1, cf. Figure 1: Exercise improves the functional and structural recovery after stroke in wild-type mice), we initially operated on 23 animals. Two mice died during surgery and one mouse (no ex) after one week. Additionally, two mice showed non-compliance in the adhesive tape removal test and were excluded from the analysis by a blinded examiner. Consequently, 18 animals were included in the final analysis of Figure 1b. For the MRI examination of infarct volume (Figure 1c) and fiber tracking (Figure 1i), we scanned seven animals per group. The tracer injections (Figure 1e-g) were performed in a separate cohort to avoid potential effects of stereotactic tracer injection on functional test results and DTI fiber tracking. In this cohort, 24 animals were initially included, of which one died during surgery, and in two cases the tracer did not reach the target area (1 ex, 1 no ex).

2) For the experiment assessing the importance of T-lymphocyte (T-cell) subsets in exercise-induced stroke recovery (Figure 2), we initially operated on different groups of animals. For the experiment comparing RAG exercise versus RAG no exercise (Figure 2b), 12 animals were operated on, with one mouse dying postoperatively. In the experiment comparing RAG + vehicle + exercise versus RAG + CD3⁺-T cell transfer + exercise (Figure 2c), 25 animals were operated on, with no mortality or exclusions. For the experiment comparing RAG + CD4⁺-T cell transfer + exercise versus RAG + CD8⁺-T cell transfer + exercise (Figure 2d), 19 animals were

operated on, with one mouse dying on postoperative day 1. The varying group sizes in this experiment resulted from differences in CD4⁺- and CD8⁺-T cell yields. Similarly, for the experiment comparing RAG + FoxP3⁺-Treg + exercise versus RAG + FoxP3⁻-Treg + exercise (Figure 2e), 17 animals were operated on, with one mouse dying on the first postoperative day. The differing group sizes were due to variations in the yield of FoxP3⁺- and FoxP3⁻-Tregs. In the experiment comparing wild-type + isotype antibody + exercise versus wild-type + anti-VLA4 antibody + exercise (Figure 2f), 14 animals were operated on, with no mortality or exclusions. Finally, for the MRI-DTI fiber tracking experiments (Figure 2g-h), 18 animals were operated on, with no mortality or exclusions. The different group sizes resulted from a limited yield of Tregs, allowing transfer in only eight animals.

In further experiments, performed to analyze the time-course of brain infiltration by Tregs in exercised mice post stroke, 9 animals were operated on, with no mortality or exclusions.

In experiments investigating the effects of either inhibiting immune cell infiltration through VLA-4 blockade in wild-type mice or Treg transfer in RAG mice, a total of 24 animals underwent surgery, of which one wild-type mouse (scheduled for isotype antibody treatment) died during the procedure.

3) For the FACS analyses of brain parenchyma, cervical lymph nodes, and spleen 14 days after photothrombotic stroke (Figure 3), 23 animals were initially operated on. One mouse died on postoperative day 8, and no animals were excluded from the analysis.

4-5) For the immunohistochemical analyses on day 49 (Figures 4 and 5), the same cohort used for the ATR analysis in Figure 1b was analyzed. Initially, 23 animals were operated on, two of which died during surgery and one after one week. Consequently, 20 animals were included in the final analysis of Figure 4. For the analyses conducted after 14 days (Figures 4 and 5), a separate cohort of 12 animals was operated on, with no mortality or exclusions.

6) For the electrophysiological analyses on day 14, 24 wild-type (wt) mice underwent surgery. One of these animals (assigned to the exercise group) died on the first postoperative day. In total, 30 RAG mice were operated on, of which three died during surgery and one on the first postoperative day (two each assigned to the stroke-only and stroke + exercise groups).

Additionally, a total of 48 sham-operated wild-type mice and 21 sham-operated RAG mice were included in the experiments. None of the sham-operated animals died.

7) For the experiment comparing RAG + Treg + exercise versus RAG + dysfunctional FoxP3-deficient CD25⁺CD4⁺ T cells (Figure 7b), 14 animals were operated on, with no mortality or exclusions. For the experiment comparing RAG + Treg + exercise versus RAG + IL-10⁻ Treg + exercise (Figure 7c), 13 animals were operated on, with no mortality or exclusions. From this cohort, 5 animals per group were included in the MRI analyses (Figure 7d-e).

Across all experiments, stroke was induced in 320 animals. Of these, 18 mice died - 9 during surgery and 9 in the postoperative period. The overall mortality rate was 5.6%. Additionally, a total of 69 sham-operated mice were included in the experiments. None of the sham-operated animals died. We have included this information in the revised manuscript.

Reviewer #4 (Remarks to the Author):

- 1 Liesz, A. *et al.* Regulatory T cells are key cerebroprotective immunomodulators in acute experimental stroke. *Nat Med* **15**, 192-199 (2009). <https://doi.org/10.1038/nm.1927>
- 2 Gliem, M., Schwaninger, M. & Jander, S. Protective features of peripheral monocytes/macrophages in stroke. *Biochim Biophys Acta* **1862**, 329-338 (2016). <https://doi.org/10.1016/j.bbadis.2015.11.004>
- 3 Saunders, D. H. *et al.* Physical fitness training for stroke patients. *Cochrane Database Syst Rev* **3**, CD003316 (2020). <https://doi.org/10.1002/14651858.CD003316.pub7>
- 4 Reitmeir, R. *et al.* Vascular endothelial growth factor induces contralesional corticobulbar plasticity and functional neurological recovery in the ischemic brain. *Acta Neuropathol* **123**, 273-284 (2012). <https://doi.org/10.1007/s00401-011-0914-z>
- 5 Minnerup, J. *et al.* Defining mechanisms of neural plasticity after brainstem ischemia in rats. *Ann Neurol* **83**, 1003-1015 (2018). <https://doi.org/10.1002/ana.25238>

- 6 Lindau, N. T. *et al.* Rewiring of the corticospinal tract in the adult rat after unilateral stroke and anti-Nogo-A therapy. *Brain* **137**, 739-756 (2014). <https://doi.org/10.1093/brain/awt336>
- 7 Akkurt, B. H. *et al.* Vasculitis and Ischemic Stroke in Lyme Neuroborreliosis-Interventional Management Approach and Literature Review. *Brain Sci* **13** (2023). <https://doi.org/10.3390/brainsci13101388>
- 8 Bouet, V. *et al.* The adhesive removal test: a sensitive method to assess sensorimotor deficits in mice. *Nat Protoc* **4**, 1560-1564 (2009). <https://doi.org/10.1038/nprot.2009.125>
- 9 Ruan, J. & Yao, Y. Behavioral tests in rodent models of stroke. *Brain Hemorrhages* **1**, 171-184 (2020). <https://doi.org/10.1016/j.heest.2020.09.001>
- 10 Scala, F. *et al.* Phenotypic variation of transcriptomic cell types in mouse motor cortex. *Nature* **598**, 144-150 (2021). <https://doi.org/10.1038/s41586-020-2907-3>
- 11 Scala, F. *et al.* Layer 4 of mouse neocortex differs in cell types and circuit organization between sensory areas. *Nat Commun* **10**, 4174 (2019). <https://doi.org/10.1038/s41467-019-12058-z>
- 12 Clarkson, A. N., Huang, B. S., Macisaac, S. E., Mody, I. & Carmichael, S. T. Reducing excessive GABA-mediated tonic inhibition promotes functional recovery after stroke. *Nature* **468**, 305-309 (2010). <https://doi.org/10.1038/nature09511>
- 13 Bice, A. R. *et al.* Homotopic contralesional excitation suppresses spontaneous circuit repair and global network reconnections following ischemic stroke. *Elife* **11** (2022). <https://doi.org/10.7554/eLife.68852>
- 14 Ellwardt, E. *et al.* Maladaptive cortical hyperactivity upon recovery from experimental autoimmune encephalomyelitis. *Nat Neurosci* **21**, 1392-1403 (2018). <https://doi.org/10.1038/s41593-018-0193-2>
- 15 Busche, M. A. *et al.* Rescue of long-range circuit dysfunction in Alzheimer's disease models. *Nat Neurosci* **18**, 1623-1630 (2015). <https://doi.org/10.1038/nn.4137>
- 16 Busche, M. A. & Konnerth, A. Neuronal hyperactivity--A key defect in Alzheimer's disease? *Bioessays* **37**, 624-632 (2015). <https://doi.org/10.1002/bies.201500004>
- 17 Putcha, D. *et al.* Hippocampal hyperactivation associated with cortical thinning in Alzheimer's disease signature regions in non-demented elderly adults. *J Neurosci* **31**, 17680-17688 (2011). <https://doi.org/10.1523/JNEUROSCI.4740-11.2011>
- 18 Ke, Z., Yip, S. P., Li, L., Zheng, X. X. & Tong, K. Y. The effects of voluntary, involuntary, and forced exercises on brain-derived neurotrophic factor and motor function recovery: a rat brain ischemia model. *PLoS One* **6**, e16643 (2011). <https://doi.org/10.1371/journal.pone.0016643>
- 19 Svensson, M. *et al.* Forced treadmill exercise can induce stress and increase neuronal damage in a mouse model of global cerebral ischemia. *Neurobiol Stress* **5**, 8-18 (2016). <https://doi.org/10.1016/j.ynstr.2016.09.002>
- 20 Curtin, N. M., Mills, K. H. & Connor, T. J. Psychological stress increases expression of IL-10 and its homolog IL-19 via beta-adrenoceptor activation: reversal by the anxiolytic chlordiazepoxide. *Brain Behav Immun* **23**, 371-379 (2009). <https://doi.org/10.1016/j.bbi.2008.12.010>
- 21 Sun, J. *et al.* Gradually increased training intensity benefits rehabilitation outcome after stroke by BDNF upregulation and stress suppression. *Biomed Res Int* **2014**, 925762 (2014). <https://doi.org/10.1155/2014/925762>
- 22 Ritzel, R. M. *et al.* Aging alters the immunological response to ischemic stroke. *Acta Neuropathol* **136**, 89-110 (2018). <https://doi.org/10.1007/s00401-018-1859-2>
- 23 Manwani, B. *et al.* Differential effects of aging and sex on stroke induced inflammation across the lifespan. *Exp Neurol* **249**, 120-131 (2013). <https://doi.org/10.1016/j.expneurol.2013.08.011>

Reviewer 1

The authors have effectively addressed my previous comments, significantly enhancing the manuscript's clarity and depth. The revised experimental design is now more detailed, and the inclusion of additional experiments strengthens the hypothesis that regulatory T cells play a role in exercise-enhanced stroke recovery. Notably, the authors have provided data demonstrating the presence of transferred Tregs in the peri-infarct region.

However, the number of transferred Tregs in this region is relatively low (<100 cells/brain), considering that 0.5 million Tregs were adoptively transferred. While this low number does not necessarily negate the potential for a localized brain effect, it raises the question of whether a peripheral effect of transferred Tregs could be modulating the systemic immune cell response, thereby affecting stroke recovery. Perhaps this point could be discussed.

Additionally, while the manuscript supports an IL-10-dependent mechanism modulating the excitability of peri-infarct neurons, it does not provide evidence of a direct IL-10 effect on neurons. To strengthen this claim, experiments that specifically block or delete the IL-10 receptor in neurons should be conducted. The authors could acknowledge this limitation in the discussion section.

Ad Reviewer 1

We sincerely thank the reviewer for the very positive assessment of our revised manuscript and for acknowledging the improvements in clarity, design, and experimental depth. We also appreciate the two remaining critical remarks, both of which we fully agree with.

In response, we have added the following two concise statements to the Discussion section of the revised manuscript:

Regarding the low number of transferred Tregs detected in the peri-infarct region, we now acknowledge that this does not exclude the possibility of systemic effects contributing to recovery:

"While the number of transferred Tregs detected in the peri-infarct region was relatively low, this does not preclude the possibility that peripheral immunomodulatory effects may also contribute to stroke recovery. We therefore acknowledge that both central and systemic mechanisms may act in concert to mediate the observed benefits of adoptively transferred Tregs."

Regarding the lack of direct evidence for neuronal IL-10 receptor signaling, we have added the following sentence to highlight this limitation:

"While our data support an IL-10-dependent mechanism modulating neuronal excitability, we cannot exclude the possibility that IL-10 may act indirectly through cell types other than

neurons. Future studies employing neuron-specific IL-10 receptor deletion or blockade will be required to confirm a direct effect of IL-10 on neurons."

We hope these additions address the reviewer's thoughtful suggestions and further improve the manuscript.

Reviewer 2

The authors have adequately addressed all the initial queries/concerns from this review.

There is one modest final concern:

The authors addressed the effects of Treg on neuronal activity in additional patch clamp experiments. However, some data have small sample sizes (Supplementary Figure 4A and B: n = 4). Since three of four samples in the Sham+ex+VTLA-4AB and RAG Stroke ipsi groups showed values far below the average of the Sham + ex group and Stroke ipsi, a larger sample size is necessary to conclude that the VLA-4 AB and RAG do not significantly change the number of stimulus-evoked APs.

Ad Reviewer 2

We thank the reviewer for the positive feedback and for acknowledging that our previous revisions have adequately addressed the initial concerns.

The remaining point regarding the limited sample size in the supplementary control experiments is entirely valid and appreciated. To address this, we have included the following sentence in the revised manuscript to appropriately reflect this limitation:

"Given the limited sample size in these control experiments, the absence of significant changes should be interpreted with some caution, though."

We hope this addition satisfactorily addresses the reviewer's final concern.